# CoinDICE: Off-Policy Confidence Interval Estimation

[*]**Bo Dai**[1], [*]    **Ofir Nachum**[1],    **Yinlam Chow**[1]
**Lihong Li**[1],    **Csaba Szepesvári**[2,3],    **Dale Schuurmans**[1,3]
[1]Google Research, Brain Team    [2]University of Alberta    [3]DeepMind

## Abstract

We study *high-confidence behavior-agnostic off-policy evaluation* in reinforcement learning, where the goal is to estimate a confidence interval on a target policy's value, given only access to a static experience dataset collected by unknown behavior policies. Starting from a function space embedding of the linear program formulation of the $Q$-function, we obtain an optimization problem with generalized estimating equation constraints. By applying the generalized empirical likelihood method to the resulting Lagrangian, we propose *CoinDICE*, a novel and efficient algorithm for computing confidence intervals. Theoretically, we prove the obtained confidence intervals are valid, in both asymptotic and finite-sample regimes. Empirically, we show in a variety of benchmarks that the confidence interval estimates are tighter and more accurate than existing methods.[2]

## 1   Introduction

One of the major barriers that hinders the application of reinforcement learning (RL) is the ability to evaluate new policies reliably *before* deployment, a problem generally known as *off-policy evaluation* (OPE). In many real-world domains, *e.g.*, healthcare (Murphy et al., 2001; Gottesman et al., 2018), recommendation (Li et al., 2011; Chen et al., 2019), and education (Mandel et al., 2014), deploying a new policy can be expensive, risky or unsafe. Accordingly, OPE has seen a recent resurgence of research interest, with many methods proposed to estimate the value of a policy (Precup et al., 2000; Dudík et al., 2011; Bottou et al., 2013; Jiang and Li, 2016; Thomas and Brunskill, 2016; Liu et al., 2018; Nachum et al., 2019a; Kallus and Uehara, 2019a,b; Zhang et al., 2020b).

However, the very settings where OPE is necessary usually entail limited data access. In these cases, obtaining knowledge of the uncertainty of the estimate is as important as having a consistent estimator. That is, rather than a *point estimate*, many applications would benefit significantly from having *confidence intervals* on the value of a policy. The problem of estimating these confidence intervals, known as *high-confidence off-policy evaluation* (HCOPE) (Thomas et al., 2015b), is imperative in real-world decision making, where deploying a policy without high-probability safety guarantees can have catastrophic consequences (Thomas, 2015). Most existing high-confidence off-policy evaluation algorithms in RL (Bottou et al., 2013; Thomas et al., 2015a,b; Hanna et al., 2017) construct such intervals using statistical techniques such as concentration inequalities and the bootstrap applied to importance corrected estimates of policy value. The primary challenge with these correction-based approaches is the high variance resulting from multiplying per-step importance ratios in long-horizon problems. Moreover, they typically require full knowledge (or a good estimate) of the behavior policy, which is not easily available in behavior-agnostic OPE settings (Nachum et al., 2019a).

In this work, we propose an algorithm for behavior-agnostic HCOPE. We start from a linear programming formulation of the state-action value function. We show that the value of the policy may be obtained from a Lagrangian optimization problem for generalized estimating equations

---

[*]Equal contribution. Email: {`bodai, ofirnachum`}@google.com.

[2]Open-source code for CoinDICE is available at https://github.com/google-research/dice_rl.

over data sampled from off-policy distributions. This observation inspires a generalized empirical likelihood approach (Owen, 2001; Broniatowski and Keziou, 2012; Duchi et al., 2016) to confidence interval estimation. These derivations enable us to express high-confidence lower and upper bounds for the policy value as results of minimax optimizations over an arbitrary offline dataset, with the appropriate distribution corrections being implicitly estimated during the optimization. åWe translate this understanding into a practical estimator, *Confidence Interval DIstribution Correction Estimation* (CoinDICE), and design an efficient algorithm for implementing it. We then justify the asymptotic coverage of these bounds and present non-asymptotic guarantees to characterize finite-sample effects. Notably, CoinDICE is behavior-agnostic and its objective function does not involve any per-step importance ratios, and so the estimator is less susceptible to high-variance gradient updates. We evaluate CoinDICE in a number of settings and show that it provides both tighter confidence interval estimates and more correctly matches the desired statistical coverage compared to existing methods.

## 2 Preliminaries

For a set $W$, the set of probability measures over $W$ is denoted by $\mathcal{P}(W)$.[3] We consider a Markov Decision Process (MDP) (Puterman, 2014), $\mathcal{M} = (S, A, T, R, \gamma, \mu_0)$, where $S$ denotes the state space, $A$ denotes the action space, $T : S \times A \to \mathcal{P}(S)$ is the transition probability kernel, $R : S \times A \to \mathcal{P}([0, R_{\max}])$ is a bounded reward kernel, $\gamma \in (0, 1]$ is the discount factor, and $\mu_0$ is the initial state distribution.

A policy, $\pi : S \to \mathcal{P}(A)$, can be used to generate a random trajectory by starting from $s_0 \sim \mu_0(s)$, then following $a_t \sim \pi(s_t)$, $r_t \sim R(s_t, a_t)$ and $s_{t+1} \sim T(s_t, a_t)$ for $t \geq 0$. The state- and action-value functions of $\pi$ are denoted $V^\pi$ and $Q^\pi$, respectively. The policy also induces an occupancy measure, $d^\pi(s, a) := (1-\gamma)\mathbb{E}_\pi\left[\sum_{t \geq 0} \gamma^t \mathbf{1}\{s_t = s, a_t = a\}\right]$, the normalized discounted probability of visiting $(s, a)$ in a trajectory generated by $\pi$, where $\mathbf{1}\{\cdot\}$ is the indicator function. Finally, the *policy value* is defined as the *normalized* expected reward accumulated along a trajectory:

$$\rho_\pi := (1-\gamma)\mathbb{E}\left[\sum_{t=0}^\infty \gamma^t r_t | s_0 \sim \mu_0, a_t \sim \pi(s_t), r_t \sim R(s_t, a_t), s_{t+1} \sim T(s_t, a_t)\right]. \quad (1)$$

We are interested in estimating the policy value and its confidence interval (CI) in the *behavior agnostic off-policy* setting (Nachum et al., 2019a; Zhang et al., 2020a), where interaction with the environment is limited to a static dataset of experience $\mathcal{D} := \{(s, a, s', r)_i\}_{i=1}^n$. Each tuple in $\mathcal{D}$ is generated according to $(s, a) \sim d^\mathcal{D}, r \sim R(s, a)$ and $s' \sim T(s, a)$, where $d^\mathcal{D}$ is an unknown distribution over $S \times A$, perhaps induced by one or more unknown behavior policies. The initial distribution $\mu_0(s)$ is assumed to be easy to sample from, as is typical in practice. Abusing notation, we denote by $d^\mathcal{D}$ both the distribution over $(s, a, s', r)$ and its marginal on $(s, a)$. We use $\mathbb{E}_d[\cdot]$ for the expectation over a given distribution $d$, and $\mathbb{E}_\mathcal{D}[\cdot]$ for its empirical approximation using $\mathcal{D}$.

Following previous work (Sutton et al., 2012; Uehara et al., 2019; Zhang et al., 2020a), for ease of exposition we assume the transitions in $\mathcal{D}$ are *i.i.d.*. However, our results may be extended to fast-mixing, ergodic MDPs, where the the empirical distribution of states along a long trajectory is close to being *i.i.d.* (Antos et al., 2008; Lazaric et al., 2012; Dai et al., 2017; Duchi et al., 2016).

Under mild regularity assumptions, the OPE problem may be formulated as a linear program – referred to as the $Q$-LP (Nachum et al., 2019b; Nachum and Dai, 2020) – with the following primal and dual forms:

$$\min_{Q: S \times A \to \mathbb{R}} (1-\gamma)\mathbb{E}_{\mu_0 \pi}[Q(s_0, a_0)] \quad (2)$$
$$\text{s.t.} \quad Q(s, a) \geq R(s, a) + \gamma \cdot \mathcal{P}^\pi Q(s, a),$$
$$\forall (s, a) \in S \times A,$$

and

$$\max_{d: S \times A \to \mathbb{R}_+} \mathbb{E}_d[r(s, a)] \quad (3)$$
$$\text{s.t.} \quad d(s, a) = (1-\gamma)\mu_0 \pi(s, a) + \gamma \cdot \mathcal{P}^\pi_* d(s, a),$$
$$\forall (s, a) \in S \times A,$$

where the operator $\mathcal{P}^\pi$ and its adjoint, $\mathcal{P}^\pi_*$, are defined as

$$\mathcal{P}^\pi Q(s, a) := \mathbb{E}_{s' \sim T(\cdot|s,a), a' \sim \pi(\cdot|s')}[Q(s', a')],$$
$$\mathcal{P}^\pi_* d(s, a) := \pi(a|s)\sum_{\tilde{s}, \tilde{a}} T(s|\tilde{s}, \tilde{a}) d(\tilde{s}, \tilde{a}).$$

The optimal solutions of (2) and (3) are the $Q$-function, $Q^\pi$, and stationary state-action occupancy, $d^\pi$, respectively, for policy $\pi$; see Nachum et al. (2019b, Theorems 3 & 5) for details as well as extensions to the undiscounted case.

Using the Lagrangian of (2) or (3), we have

$$\rho_\pi = \min_Q \max_{\tau \geqslant 0} \ (1-\gamma)\,\mathbb{E}_{\mu_0\pi}\left[Q\left(s_0, a_0\right)\right] + \mathbb{E}_{d^\mathcal{D}}\left[\tau\left(s,a\right)\left(R\left(s,a\right)+\gamma Q\left(s',a'\right)-Q\left(s,a\right)\right)\right], \quad (4)$$

where $\tau\left(s,a\right) := \frac{d(s,a)}{d^\mathcal{D}(s,a)}$ is the *stationary distribution corrector*. One of the key benefits of the minimax optimization (4) is that both expectations can be immediately approximated by sample averages.[4] In fact, this formulation allows the derivation of several recent behavior-agnostic OPE estimators in a unified manner (Nachum et al., 2019a; Uehara et al., 2019; Zhang et al., 2020a; Nachum and Dai, 2020).

# 3 CoinDICE

We now develop a new approach to obtaining confidence intervals for OPE. The algorithm, *COnfidence INterval stationary DIstribution Correction Estimation (CoinDICE)*, is derived by combining function space embedding and the previously described $Q$-LP.

## 3.1 Function Space Embedding of Constraints

Both the primal and dual forms of the $Q$-LP contain $|S||A|$ constraints that involve expectations over state transition probabilities. Working directly with these constraints quickly becomes computationally and statistically prohibitive when $|S||A|$ is large or infinite, as with standard LP approaches (De Farias and Van Roy, 2003). Instead, we consider a relaxation that embeds the constraints in a function space:

$$\tilde{\rho}_\pi := \max_{d:S\times A \to \mathbb{R}_+} \mathbb{E}_d\left[r\left(s,a\right)\right] \quad \text{s.t.} \ \langle \phi, d \rangle = \langle \phi, (1-\gamma)\,\mu_0\pi + \gamma \cdot \mathcal{P}_*^\pi d \rangle, \quad (5)$$

where $\phi : S \times A \to \Omega^p \subset \mathbb{R}^p$ is a feature map, and $\langle \phi, d \rangle := \int \phi\left(s,a\right) d\left(s,a\right) ds da$. By projecting the constraints onto a function space with feature mapping $\phi$, we can reduce the number of constraints from $|S||A|$ to $p$. Note that $p$ may still be infinite. The constraint in (5) can be written as *generalized estimating equations* (Qin and Lawless, 1994; Lam and Zhou, 2017) for the correction ratio $\tau\left(s,a\right)$ over augmented samples $x := (s_0, a_0, s, a, r, s', a')$ with $(s_0, a_0) \sim \mu_0\pi$, $(s, a, r, s') \sim d^\mathcal{D}$, and $a' \sim \pi(\cdot|s')$,

$$\langle \phi, d \rangle = \langle \phi, (1-\gamma)\,\mu_0\pi + \gamma \cdot \mathcal{P}_*^\pi d \rangle \quad \Leftrightarrow \quad \mathbb{E}_x\left[\Delta\left(x;\tau,\phi\right)\right] = 0, \quad (6)$$

where $\Delta\left(x;\tau,\phi\right) := (1-\gamma)\,\phi\left(s_0, a_0\right) + \tau\left(s,a\right)\left(\gamma\phi\left(s',a'\right) - \phi\left(s,a\right)\right)$. The corresponding Lagrangian is

$$\tilde{\rho}_\pi = \max_{\tau:S\times A\to\mathbb{R}_+} \min_{\beta\in\mathbb{R}^p} \mathbb{E}_{d^\mathcal{D}}\left[\tau \cdot r\left(s,a\right)\right] + \langle \beta, \mathbb{E}_{d^\mathcal{D}}\left[\Delta\left(x;\tau,\phi\right)\right]\rangle. \quad (7)$$

This embedding approach for the dual $Q$-LP is closely related to approximation methods for the standard state-value LP (De Farias and Van Roy, 2003; Pazis and Parr, 2011; Lakshminarayanan et al., 2017). The gap between the solutions to (5) and the original dual LP (3) depends on the expressiveness of the feature mapping $\phi$. Before stating a theorem that quantifies the error, we first offer a few examples to provide intuition for the role played by $\phi$.

**Example (Indicator functions):** Suppose $p = |S||A|$ is finite and $\phi = [\delta_{s,a}]_{(s,a)\in S\times A}$, where $\delta_{s,a} \in \{0,1\}^p$ with $\delta_{s,a} = 1$ at position $(s,a)$ and 0 otherwise. Plugging this feature mapping into (5), we recover the original dual $Q$-LP (3).

**Example (Full-rank basis):** Suppose $\Phi \in \mathbb{R}^{p\times p}$ is a full-rank matrix with $p = |S||A|$; furthermore, $\phi(s,a) = \Phi((s,a), \cdot)^\top$. Although the constraints in (5) and (3) are different, their solutions are identical. This can be verified by the Lagrangian in Appendix A.

**Example (RKHS function mappings):** Suppose $\phi\left(s,a\right) := k\left((s,a),\cdot\right) \in \mathbb{R}^p$ with $p = \infty$, which forms a reproducing kernel Hilbert space (RKHS) $\mathcal{H}_k$. The LHS and RHS in the constraint of (5) are the kernel embeddings of $d\left(s,a\right)$ and $(1-\gamma)\,\mu_0\pi\left(s,a\right) + \gamma \cdot \mathcal{P}_*^\pi d\left(s,a\right)$ respectively. The constraint in (5) can then be understood as as a form of distribution matching by comparing

kernel embeddings, rather than element-wise matching as in (3). If the kernel function $k(\cdot, \cdot)$ is characteristic, the embeddings of two distributions will match if and only if the distributions are identical almost surely (Sriperumbudur et al., 2011).

**Theorem 1 (Approximation error)** *Suppose the constant function $\mathbf{1} \in \mathcal{F}_\phi := \mathrm{span}\{\phi\}$. Then,*

$$0 \leqslant \tilde{\rho}_\pi - \rho_\pi \leqslant 2 \min_\beta \|Q^\pi - \langle \beta, \phi \rangle\|_\infty,$$

*where $Q^\pi$ is the fixed-point solution to the Bellman equation $Q(s,a) = R(s,a) + \gamma \mathcal{P}^\pi Q(s,a)$.*

Please refer to Appendix A for the proof. The condition $\mathbf{1} \in \mathcal{F}_\phi$ is standard and is trivial to satisfy. Although the approximation error relies on $\|\cdot\|_\infty$, a sharper bound that relies on a norm taking the state-action distribution into account can also be obtained (De Farias and Van Roy, 2003). We focus on characterizing the uncertainty due to sampling in this paper, so for ease of exposition we will consider a setting where $\phi$ is sufficiently expressive to make the approximation error zero. If desired, the approximation error in Theorem 1 can be included in the analysis.

Note that, compared to using a characteristic kernel to ensure injectivity for the RKHS embeddings over all distributions (and thus guaranteeing arbitrarily small approximation error), Theorem 1 only requires that $Q^\pi$ be represented in $\mathcal{F}_\phi$, which is a much weaker condition. In practice, one may also learn the feature mapping $\phi$ for the projection jointly.

## 3.2 Off-policy Confidence Interval Estimation

By introducing the function space embedding of the constraints in (5), we have transformed the original point-wise constraints in the $Q$-LP to generalized estimating equations. This paves the way to applying the generalized empirical likelihood (EL) (Owen, 2001; Broniatowski and Keziou, 2012; Bertail et al., 2014; Duchi et al., 2016) method to estimate a confidence interval on policy value.

Recall that, given a convex, lower-semicontinuous function $f : \mathbb{R}_+ \to \mathbb{R}$ satisfying $f(1) = 0$, the $f$-divergence between densities $p$ and $q$ on $\mathbb{R}$ is defined as $D_f(P\|Q) := \int Q(dx) f\left(\frac{dP(x)}{dQ(x)}\right) dx$.

Given an $f$-divergence, we propose our main confidence interval estimate based on the following confidence set $C_{n,\xi}^f \subset \mathbb{R}$:

$$C_{n,\xi}^f := \left\{ \tilde{\rho}_\pi(w) = \max_{\tau \geqslant 0} \mathbb{E}_w[\tau \cdot r] \,\middle|\, w \in \mathcal{K}_f, \mathbb{E}_w[\Delta(x; \tau, \phi)] = 0 \right\} \text{ with } \mathcal{K}_f := \left\{ \begin{array}{l} w \in \mathcal{P}^{n-1}(\widehat{p}_n), \\ D_f(w\|\widehat{p}_n) \leqslant \frac{\xi}{n} \end{array} \right\},$$
(8)

where $\mathcal{P}^{n-1}(\widehat{p}_n)$ denotes the $n$-simplex on the support of $\widehat{p}_n$, the empirical distribution over $\mathcal{D}$. It is easy to verify that this set $C_{n,\xi}^f \subset \mathbb{R}$ is convex, since $\tilde{\rho}_\pi(w)$ is a convex function over a convex feasible set. Thus, $C_{n,\xi}^f$ is an interval. In fact, $C_{n,\xi}^f$ is the image of the policy value $\tilde{\rho}_\pi$ on a bounded (in $f$-divergence) perturbation to $w$ in the neighborhood of the empirical distribution $\widehat{p}_n$.

Intuitively, the confidence interval $C_{n,\xi}^f$ possesses a close relationship to bootstrap estimators. In vanilla bootstrap, one constructs a set of empirical distributions $\{w^i\}_{i=1}^m$ by resampling from the dataset $\mathcal{D}$. Such subsamples are used to form the empirical distribution on $\{\tilde{\rho}(w^i)\}_{i=1}^m$, which provides population statistics for confidence interval estimation. However, this procedure is computationally very expensive, involving $m$ separate optimizations. By contrast, our proposed estimator $C_{n,\xi}^f$ exploits the asymptotic properties of the statistic $\tilde{\rho}_\pi(w)$ to derive a target confidence interval by solving only *two* optimization problems (Section 3.3), a dramatic savings in computational cost.

Before introducing the algorithm for computing $C_{n,\xi}^f$, we establish the first key result that, by choosing $\xi = \chi_{(1)}^{2,1-\alpha}$, $C_{n,\xi}^f$ is asymptotically a $(1-\alpha)$-confidence interval on the policy value, where $\chi_{(1)}^{2,1-\alpha}$ is the $(1-\alpha)$-quantile of the $\chi^2$-distribution with 1 degree of freedom.

**Theorem 2 (Informal asymptotic coverage)** *Under some mild conditions, if $\mathcal{D}$ contains i.i.d. samples and the optimal solution to the Lagrangian of (5) is unique, we have*

$$\lim_{n \to \infty} \mathbb{P}\left(\rho_\pi \in C_{n,\xi}^f\right) = \mathbb{P}\left(\chi_{(1)}^2 \leqslant \xi\right).$$
(9)

*Thus, $C^f_{n,\chi^{2,1-\alpha}_{(1)}}$ is an asymptotic $(1-\alpha)$-confidence interval of the value of the policy $\pi$.*

Please refer to Appendix E.1 for the precise statement and proof of Theorem 2.

Theorem 2 generalizes the result in Duchi et al. (2016) to statistics with generalized estimating equations, maintaining the 1 degree of freedom in the asymptotic $\chi^2_{(1)}$-distribution. One may also apply existing results for EL with generalized estimating equations (e.g., Lam and Zhou, 2017), but these would lead to a limiting distribution of $\chi^2_{(m)}$ with $m \gg 1$ degrees of freedom, resulting in a much looser confidence interval estimate than Theorem 2.

Note that Theorem 2 can also be specialized to multi-armed contextual bandits to achieve a tighter confidence interval estimate in this special case. In particular, for contextual bandits, the stationary distribution constraint in (5), $\mathbb{E}_w\left[\Delta\left(x;\tau,\phi\right)\right] = 0$, is no longer needed, and can be replaced by $\mathbb{E}_w\left[\tau - 1\right] = 0$. Then by the same technique used for MDPs, we can obtain a confidence interval estimate for offline contextual bandits; see details in Appendix C. Interestingly, the resulting confidence interval estimate not only has the same asymptotic coverage as previous work (Karampatziakis et al., 2019), but is also simpler and computationally more efficient.

### 3.3 Computing the Confidence Interval

Now we provide a distributional robust optimization view of the upper and lower bounds of $C^f_{n,\xi}$.

**Theorem 3 (Upper and lower confidence bounds)** *Denote the upper and lower confidence bounds of $C^f_{n,\xi}$ by $u_n$ and $l_n$, respectively:*

$$[l_n, u_n] = \left[\min_{w \in \mathcal{K}_f} \min_{\beta \in \mathbb{R}^p} \max_{\tau \geqslant 0} \mathbb{E}_w\left[\ell\left(x;\tau,\beta\right)\right], \quad \max_{w \in \mathcal{K}_f} \max_{\tau \geqslant 0} \min_{\beta \in \mathbb{R}^p} \mathbb{E}_w\left[\ell\left(x;\tau,\beta\right)\right]\right], \quad (10)$$

$$= \left[\min_{\beta \in \mathbb{R}^p} \max_{\tau \geqslant 0} \min_{w \in \mathcal{K}_f} \mathbb{E}_w\left[\ell\left(x;\tau,\beta\right)\right], \quad \max_{\tau \geqslant 0} \min_{\beta \in \mathbb{R}^p} \max_{w \in \mathcal{K}_f} \mathbb{E}_w\left[\ell\left(x;\tau,\beta\right)\right]\right], \quad (11)$$

*where $\ell\left(x;\tau,\beta\right) := \tau \cdot r + \beta^\top \Delta\left(x;\tau,\phi\right)$. For any $(\tau,\beta,\lambda,\eta)$ that satisfies the constraints in (11), the optimal weights for the upper and lower confidence bounds are*

$$w_l = f'_*\left(\frac{\eta - \ell\left(x;\tau,\beta\right)}{\lambda}\right) \quad and \quad w_u = f'_*\left(\frac{\ell\left(x;\tau,\beta\right) - \eta}{\lambda}\right). \quad (12)$$

*respectively. Therefore, the confidence bounds can be simplified as:*

$$\begin{bmatrix} l_n \\ u_n \end{bmatrix} = \begin{bmatrix} \min_\beta \max_{\tau \geqslant 0, \lambda \geqslant 0, \eta} \mathbb{E}_{\mathcal{D}}\left[-\lambda f_*\left(\frac{\eta - \ell(x;\tau,\beta)}{\lambda}\right) + \eta - \lambda\frac{\xi}{n}\right] \\ \max_{\tau \geqslant 0} \min_{\beta, \lambda \geqslant 0, \eta} \mathbb{E}_{\mathcal{D}}\left[\lambda f_*\left(\frac{\ell(x;\tau,\beta) - \eta}{\lambda}\right) + \eta + \lambda\frac{\xi}{n}\right] \end{bmatrix}. \quad (13)$$

The proof of this result relies on Lagrangian duality and the convexity and concavity of the optimization; it may be found in full detail in Appendix D.1.

As we can see in Theorem 3, by exploiting strong duality properties to move $w$ into the inner most optimizations in (11), the obtained optimization (11) is the distributional robust optimization extenion of the saddle-point problem. The closed-form reweighting scheme is demonstrated in (12). For particular $f$-divergences, such as the $KL$- and 2-power divergences, for a fixed $(\beta, \tau)$, the optimal $\eta$ can be easily computed and the weights $w$ recovered in closed-form. For example, by using $KL\left(w||\widehat{p}_n\right)$, (12) can be used to obtain the updates

$$w_l\left(x\right) = \exp\left(\frac{\eta_l - \ell\left(x;\tau,\beta\right)}{\lambda}\right), \quad w_u\left(x\right) = \exp\left(\frac{\ell\left(x;\tau,\beta\right) - \eta_u}{\lambda}\right), \quad (14)$$

where $\eta_l$ and $\eta_u$ provide the normalizing constants. (For closed-form updates of $w$ w.r.t. other $f$-divergences, please refer to Appendix D.2.) Plug the closed-form of optimal weights into (11), this greatly simplifies the optimization over the data perturbations yielding (13), and estabilishes the connection to the prioritized experiences replay (Schaul et al., 2016), where both reweight the experience data according to their loss, but with different reweighting schemes.

Note that it is straightforward to check that the estimator for $u_n$ in (13) is nonconvex-concave and the estimator for $l_n$ in (13) is nonconcave-convex. Therefore, one could alternatively apply stochastic gradient descent-ascent (SGDA) for to solve (13) and benefit from attractive finite-step convergence guarantees (Lin et al., 2019).

**Remark (Practical considerations):** As also observed in Namkoong and Duchi (2016), SGDA for (13) could potentially suffer from high variance in both the objective and gradients when $\lambda$ approaches 0. In Appendix D.3, we exploit several properties of (11), which leads to a computational efficient algorithm, to overcome the numerical issue. Please refer to Appendix D.3 for the details of Algorithm 1 and the practical considerations.

**Remark (Joint learning for feature embeddings):** The proposed framework also allows for the possibility to learn the features for constraint projection. In particular, consider $\zeta(\cdot, \cdot) := \beta^\top \phi(\cdot, \cdot) : S \times A \to \mathbb{R}$. Note that we could treat the combination $\beta^\top \phi(s, a)$ together as the Lagrange multiplier function for the original $Q$-LP with *infinitely* many constraints, hence both $\beta$ and $\phi(\cdot, \cdot)$ could be updated jointly. Although the conditions for asymptotic coverage no longer hold, the finite-sample correction results of the next section are still applicable. This might offer an interesting way to reduce the approximation error introduced by inappropriate feature embeddings of the constraints, while still maintaining calibrated confidence intervals.

## 4 Finite-sample Analysis

Theorem 2 establishes the asymptotic $(1 - \alpha)$-coverage of the confidence interval estimates produced by CoinDICE, ignoring higher-order error terms that vanish as sample size $n \to \infty$. In practice, however, $n$ is always finite, so it is important to quantify these higher-order terms. This section addresses this problem, and presents a finite-sample bound for the estimate of CoinDICE. In the following, we let $\mathcal{F}_\tau$ and $\mathcal{F}_\beta$ be the function classes of $\tau$ and $\beta$ used by CoinDICE.

**Theorem 4 (Informal finite-sample correction)** *Denote by $d_{\mathcal{F}_\tau}$ and $d_{\mathcal{F}_\beta}$ the finite VC-dimension of $\mathcal{F}_\tau$ and $\mathcal{F}_\beta$, respectively. Under some mild conditions, when $D_f$ is $\chi^2$-divergence, we have*

$$\mathbb{P}\left(\rho_\pi \in [l_n - \kappa_n, u_n + \kappa_n]\right) \geqslant 1 - 12 \exp\left(c_1 + 2\left(d_{\mathcal{F}_\tau} + d_{\mathcal{F}_\beta} - 1\right)\log n - \frac{\xi}{18}\right),$$

*where $c_1 = 2c + \log d_{\mathcal{F}_\tau} + \log d_{\mathcal{F}_\beta} + \left(d_{\mathcal{F}_\tau} + d_{\mathcal{F}_\beta} - 1\right)$, $\kappa_n = \frac{11 M \xi}{6n} + 2\frac{C_\ell M}{n}\left(1 + 2\sqrt{\frac{\xi}{9n}}\right)$, and $(c, M, C_\ell)$ are univeral constants.*

The precise statement and detailed proof of Theorem 4 can be found in Appendix E.2. The proof relies on empirical Bernstein bounds with a careful analysis of the variance term. Compared to the vanilla sample complexity of $\mathcal{O}\left(\frac{1}{\sqrt{n}}\right)$, we achieve a faster rate of $\mathcal{O}\left(\frac{1}{n}\right)$ without any additional assumptions on the noise or curvature conditions. The tight sample complexity in Theorem 4 implies that one can construct the $(1 - \alpha)$-finite sample confidence interval by optimizing (11) with $\xi = 18\left(\log\frac{\alpha}{12} - c_1 - 2\left(d_{\mathcal{F}_\tau} + d_{\mathcal{F}_\beta} - 1\right)\log n\right)$, and composing with $\kappa_n$. However, we observe that this bound can be conservative compared to the asymptotic confidence interval in Theorem 2. Therefore, we will evaluate the asymptotic version of CoinDICE based on Theorem 2 in the experiment.

The conservativeness arises from the use of a union bound. However, we conjecture that the rate is optimal up to a constant. We exploit the VC dimension due to its generality. In fact, the bound can be improved by considering a data-dependent measure, *e.g.*, Rademacher complexity, or by some function class dependent measure, *e.g.*, function norm in RKHS, for specific function approximators.

## 5 Optimism vs. Pessimism Principle

CoinDICE provide both upper and lower bounds of the target policy's estimated value, which paves the path for applying the principle of optimism (Lattimore and Szepesvári, 2020) or pessimism (Swaminathan and Joachims, 2015) in the face of uncertainty for policy optimization in different learning settings.

**Optimism in the face of uncertainty.** Optimism in the face of uncertainty leads to *risk-seeking* algorithms, which can be used to balance the exploration/exploitation trade-off. Conceptually, they always treat the environment as the best plausibly possible. This principle has been successfully applied to stochastic bandit problems, leading to many instantiations of UCB algorithms (Lattimore and Szepesvári, 2020). In each round, an action is selected according to the upper confidence bound, and the obtained reward will be used to refine the confidence bound iteratively. When applied to MDPs, this principle inspires many optimistic model-based (Bartlett and Mendelson, 2002; Auer

et al., 2009; Strehl et al., 2009; Szita and Szepesvari, 2010; Dann et al., 2017), value-based (Jin et al., 2018), and policy-based algorithms (Cai et al., 2019). Most of these algorithms are not compatible with function approximators.

We can also implement the optimism principle by optimizing the upper bound in CoinDICE iteratively, *i.e.*, $\max_\pi u_\mathcal{D}(\pi)$. In $t$-th iteration, we calculate the gradient of $u_\mathcal{D}(\pi^t)$, *i.e.*, $\nabla_\pi u_\mathcal{D}(\pi^t)$, based on the existing dataset $\mathcal{D}_t$, then, the policy $\pi_t$ will be updated by (natural) policy gradient and samples will be collected through the updated policy $\pi_{t+1}$. Please refer to Appendix F for the gradient computation and algorithm details.

**Pessimism in the face of uncertainty.** In offline reinforcement learning (Lange et al., 2012; Fujimoto et al., 2019; Wu et al., 2019; Nachum et al., 2019b), only a fixed set of data from behavior policies is given, a safe optimization criterion is to maximize the worst-case performance among a set of statistically plausible models (Laroche et al., 2019; Kumar et al., 2019; Yu et al., 2020). In contrast to the previous case of online exploration, this is a pessimism principle (Cohen and Hutter, 2020; Buckman et al., 2020) or counterfactual risk minimization (Swaminathan and Joachims, 2015), and highly related to robust MDP (Iyengar, 2005; Nilim and El Ghaoui, 2005; Tamar et al., 2013; Chow et al., 2015).

Different from most of the existing methods where the worst-case performance is characterized by model-based perturbation or ensemble, the proposed CoinDICE provides a lower bound to implement the pessimism principle, *i.e.*, $\max_\pi l_\mathcal{D}(\pi)$. Conceptually, we apply the (natural) policy gradient w.r.t. $l_\mathcal{D}(\pi^t)$ to update the policy iteratively. Since we are dealing with policy optimization in the offline setting, the dataset $\mathcal{D}$ keeps unchanged. Please refer to Appendix F for the algorithm details.

# 6 Related Work

Off-policy estimation has been extensively studied in the literature, given its practical importance. Most existing methods are based on the core idea of mportance reweighting to correct for distribution mismatches between the target policy and the off-policy data (Precup et al., 2000; Bottou et al., 2013; Li et al., 2015; Xie et al., 2019). Unfortunately, when applied naively, importance reweighting can result in an excessively high variance, which is known as the "curse of horizon" (Liu et al., 2018). To avoid this drawback, there has been rapidly growing interest in estimating the correction ratio of the *stationary* distribution (e.g., Liu et al., 2018; Nachum et al., 2019a; Uehara et al., 2019; Liu et al., 2019; Zhang et al., 2020a,b). This work is along the same line and thus applicable in long-horizon problems. Other off-policy approaches are also possible, notably model-based (e.g., Fonteneau et al., 2013) and doubly robust methods (Jiang and Li, 2016; Thomas and Brunskill, 2016; Tang et al., 2020; Uehara et al., 2019). These techniques can potentially be combined with our algorithm, which we leave for future investigation.

While most OPE works focus on obtaining accurate *point* estimates, several authors provide ways to quantify the amount of uncertainty in the OPE estimates. In particular, confidence bounds have been developed using the central limit theorem (Bottou et al., 2013), concentration inequalities (Thomas et al., 2015b; Kuzborskij et al., 2020), and nonparametric methods such as the bootstrap (Thomas et al., 2015a; Hanna et al., 2017). In contrast to these works, the CoinDICE is asymptotically pivotal, meaning that there are no hidden quantities we need to estimate, which is based on correcting for the stationary distribution in the *behavior-agnostic* setting, thus avoiding the curse of horizon and broadening the application of the uncertainty estimator. Recently, Jiang and Huang (2020) provide confidence intervals for OPE, but focus on the intervals determined by the *approximation error* induced by a function approximator, while our confidence intervals quantify *statistical error*.

Empirical likelihood (Owen, 2001) is a powerful tool with many applications in statistical inference like econometrics (Chen et al., 2018), and more recently in distributionally robust optimization (Duchi et al., 2016; Lam and Zhou, 2017). EL-based confidence intervals can be used to guide exploration in multiarmed bandits (Honda and Takemura, 2010; Cappé et al., 2013), and for OPE (Karampatziakis et al., 2019; Kallus and Uehara, 2019b). While the work of Kallus and Uehara (2019b) is also based on EL, it differs from the present work in two important ways. First, their focus is on developing an asymptotically efficient OPE *point* estimate, not confidence intervals. Second, they solve for timestep-dependent weights, whereas we only need to solve for timestep-*independent* weights from a system of moment matching equations induced by an underlying ergodic Markov chain.

# 7 Experiments

We now evaluate the empirical performance of CoinDICE, comparing it to a number of existing confidence interval estimators for OPE based on concentration inequalities. Specifically, given a dataset of logged trajectories, we first use weighted step-wise importance sampling (Precup et al., 2000) to calculate a separate estimate of the target policy value for each trajectory. Then given such a finite sample of estimates, we then use the empirical *Bernstein* inequality (Thomas et al., 2015b) to derive high-confidence lower and upper bounds for the true value. Alternatively, one may also use *Student's t-test* or Efron's bias corrected and accelerated *bootstrap* (Thomas et al., 2015a).

We begin with a simple bandit setting, devising a two-armed bandit problem with stochastic payoffs. We define the target policy as a near-optimal policy, which chooses the optimal arm with probability 0.95. We collect off-policy data using a behavior policy which chooses the optimal arm with probability of only 0.55. Our results are presented in Figure 1. We plot the empirical coverage and width of the estimated intervals across different confidence levels. More specifically, each data point in Figure 1 is the result of 200 experiments. In each experiment, we randomly sample a dataset and then compute a confidence interval. The *interval coverage* is then computed as the proportion of intervals out of 200 that contain the true value of the target policy. The *interval log-width* is the median of the log of the width of the 200 computed intervals. Figure 1 shows that the intervals

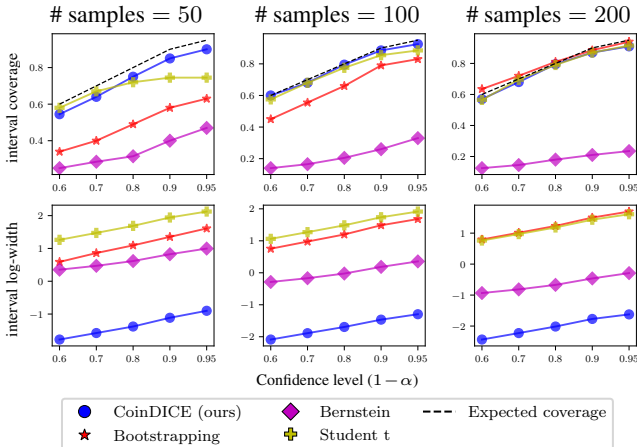

Figure 1: Results of CoinDICE and baseline methods on a simple two-armed bandit. We plot empirical coverage and median log-width ($y$-axes) of intervals evaluated at a number of desired confidence levels ($x$-axis), as measured over 200 random trials. We find that CoinDICE achieves more accurate coverage and narrower intervals compared to the baseline confidence interval estimation methods.

produced by CoinDICE achieve an empirical coverage close to the intended coverage. In this simple bandit setting, the coverages of Student's $t$ and bootstrapping are also close to correct, although they suffer more in the low-data regime. Notably, the width of the intervals produced by CoinDICE are especially narrow while maintaining accurate coverage.

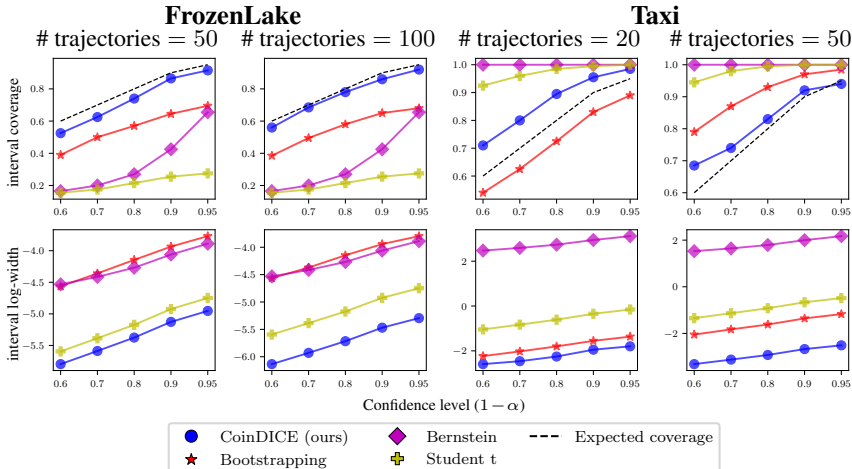

Figure 2: Results of CoinDICE and baseline methods on an infinite-horizon version of FrozenLake and Taxi. In FrozenLake, each dataset consists of trajectories of length 100; in Taxi, each dataset consists of trajectories of length 500.

We now turn to more complicated MDP environments. We use FrozenLake (Brockman et al., 2016), a highly stochastic gridworld environment, and Taxi (Dietterich, 1998), an environment with a moderate state space of 2 000 elements. As in (Liu et al., 2018), we modify these environments to be infinite horizon by randomly resetting the state upon termination. The discount factor is $\gamma = 0.99$. The target policy is taken to be a near-optimal one, while the behavior policy is highly suboptimal. The behavior policy in FrozenLake is the optimal policy with 0.2 white noise, which reduces the policy value dramatically, from 0.74 to 0.24. For the behavior policies in Taxi and Reacher, we follow the same experiment setting for constructing the behavior policies to collect data as in (Nachum et al., 2019a; Liu et al., 2018).

We follow the same evaluation protocol as in the bandit setting, measuring empirical interval coverage and log-width over 200 experimental trials for various dataset sizes and confidence levels. Results are shown in Figure 2. We find a similar conclusion that CoinDICE consistently achieves more accurate coverage and smaller widths than baselines. Notably, the baseline methods' accuracy suffers more significantly compared to the simpler bandit setting described earlier.

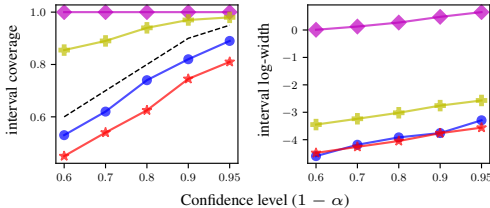

Figure 3: Results of CoinDICE and baseline methods on Reacher (Brockman et al., 2016; Todorov et al., 2012), using 25 trajectories of length 100. Colors and markers are as defined in the legends of previous figures.

Lastly, we evaluate CoinDICE on Reacher (Brockman et al., 2016; Todorov et al., 2012), a continuous control environment. In this setting, we use a one-hidden-layer neural network with ReLU activations. Results are shown in Figure 3. To account for the approximation error of the used neural network, we measure the coverage of CoinDICE with respect to a true value computed as the median of a large ensemble of neural networks trained on the off-policy data. To keep the comparison fair, we measure the coverage of the IS-based baselines with respect to a true value computed as the median of a large number of IS-based point estimates. The results show similar conclusions as before: CoinDICE achieves more accurate coverage than the IS-based methods. Still, we see that CoinDICE coverage suffers in this regime, likely due to optimization difficulties. If the optimum of the Lagrangian is only approximately found, the empirical coverage will inevitably be inexact.

## 8  Conclusion

In this paper, we have developed CoinDICE, a novel and efficient confidence interval estimator applicable to the *behavior-agnostic offline* setting. The algorithm builds on a few technical components, including a new feature embedded $Q$-LP, and a generalized empirical likelihood approach to confidence interval estimation. We analyzed the asymptotic coverage of CoinDICE's estimate, and provided an inite-sample bound. On a variety of off-policy benchmarks we empirically compared the new algorithm with several strong baselines and found it to be superior to them.

## Broader Impact

This research is fundamental and targets a broad question in reinforcement learning. The ability to reliably assess uncertainty in off-policy evaluation would have significant positive benefits for safety-critical applications of reinforcement learning. Inaccurate uncertainty estimates create the danger of misleading decision makers and could lead to detrimental consequences. However, our primary goal is to improve these estimators and reduce the ultimate risk of deploying reinforcement-learned systems. The techniques are general and do not otherwise target any specific application area.

## Acknowledgements

We thank Hanjun Dai, Mengjiao Yang and other members of the Google Brain team for helpful discussions. Csaba Szepesvári gratefully acknowledges funding from the Canada CIFAR AI Chairs Program, Amii and NSERC.

## Footnotes

[3] All sets and maps are assumed to satisfy appropriate measurability conditions; which we will omit from below for the sake of reducing clutter.

[4]We assume one can sample initial states from $\mu_0$, an assumption that often holds in practice. Then, the data in $\mathcal{D}$ can be treated as being augmented as $(s_0, a_0, s, a, r, s', a')$ with $a_0 \sim \pi\left(a|s_0\right), a' \sim \pi\left(a|s'\right)$.

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
