[Supplementary Material]

# Appendix

## A   Approximation Error Analysis

In this section, we provide a complete proof of Theorem 1, quantifying the effect of function embedding of constraints in dual $Q$-LP. The proof is an adaptation from the standard LP for state-value functions to the case of $Q$-LP (De Farias and Van Roy, 2003).

We first provide an equivalent reformulation of the primal of the feature embedded LP,

**Lemma 5** *The solution defined by*
$$\beta^* = \underset{\beta \in \mathbb{R}^p}{\operatorname{argmin}} \left\{ (1-\gamma) \mathbb{E}_{\mu_0 \pi} \left[ \beta^\top \phi(s_0, a_0) \right] | \beta^\top \phi(s,a) \geqslant \mathcal{B}_\pi \left( \beta^\top \phi \right)(s,a), \ \forall (s,a) \in S \times A \right\},$$
*with* $(\mathcal{B}_\pi Q)(s,a) := R(s,a) + \gamma \cdot \mathcal{P}^\pi Q(s,a)$ *is also the solution to*

$$\min_{\beta \in \mathbb{R}^p} \quad \left\| Q^\pi - \beta^\top \phi \right\|_{1, \mu_0 \pi} \tag{15}$$

$$\text{s.t.} \quad \beta^\top \phi(s,a) \geqslant \mathcal{B}_\pi \left( \beta^\top \phi \right)(s,a), \ \forall (s,a) \in S \times A,$$

*where* $\|f\|_{1,\mu_0 \pi} := \int |f(s,a)| \mu_0(s) \pi(a|s) \, ds \, da.$

**Proof** Recall the fact that $\mathcal{B}_\pi$ is monotonic: given two bounded functions, $\nu_1 \geqslant \nu_2$ implies $\mathcal{B}_\pi \nu_1 \geqslant \mathcal{B}_\pi \nu_2$. Therefore, for any feasible $\nu$, we have $\nu \geqslant \mathcal{B}_\pi \nu \geqslant \mathcal{B}_\pi^2 \nu \geqslant \ldots \geqslant \mathcal{B}_\pi^\infty \nu = Q^\pi$, where the convergence to $Q^\pi$ is due to the contraction property of $\mathcal{B}_\pi$.

Consider a feasible $\beta$, we have

$$\left\| Q^\pi - \beta^\top \phi \right\|_{1, \mu_0 \pi} = \int \left( \beta^\top \phi(s,a) - Q^\pi(s,a) \right) \mu_0(s) \pi(a|s) \, ds \, da, \tag{16}$$

which implies minimizing $\mathbb{E}_{\mu_0 \pi} \left[ \beta^\top \phi \right]$ is equivalent to minimizing $\left\| Q^\pi - \beta^\top \phi \right\|_{1, \mu_0 \pi}$. ∎

**Theorem 1** *Suppose the constant function* $\mathbf{1} \in \mathcal{F}_\phi := \operatorname{span}\{\phi\}$. *Then,*
$$0 \leqslant \tilde{\rho}_\pi - \rho_\pi \leqslant 2 \min_\beta \left\| Q^\pi - \langle \beta, \phi \rangle \right\|_\infty,$$

*where* $Q^\pi$ *is the fixed-point solution to the Bellman equation* $Q(s,a) = R(s,a) + \gamma \mathcal{P}^\pi Q(s,a).$

**Proof** We first show the equivalence between function space embedding of dual $Q$-LP and the linear approximation of primal $Q$-LP, which can be easily derived by checking their Lagrangians. Denote

$$l(d, \beta) := \mathbb{E}_d \left[ r(s,a) \right] + \beta^\top \langle \phi, (1-\gamma) \mu_0 \pi + \gamma \cdot \mathcal{P}_*^\pi d - d \rangle \tag{17}$$
$$= (1-\gamma) \mathbb{E}_{\mu_0 \pi} \left[ \beta^\top \phi(s,a) \right] + \mathbb{E}_d \left[ r(s,a) + \gamma \cdot \mathcal{P}^\pi \beta^\top \phi(s,a) - \beta^\top \phi(s,a) \right]$$
$$= (1-\gamma) \mathbb{E}_{\mu_0 \pi} \left[ Q_\beta(s,a) \right] + \mathbb{E}_d \left[ r(s,a) + \gamma \cdot \mathcal{P}^\pi Q_\beta(s,a) - Q_\beta(s,a) \right],$$

where $\beta \in \mathbb{R}^p$ and $Q_\beta(s,a) := \beta^\top \phi(s,a)$. Since the $l(d, \beta)$ is convex-concave w.r.t. $(\beta, d)$, it is also the Lagrangian of primal $Q$-LP with linear parametrization, *i.e.*,

$$\min_{\beta \in \mathbb{R}^p} \quad (1-\gamma) \mathbb{E}_{\mu_0 \pi} \left[ \beta^\top \phi(s_0, a_0) \right] \tag{18}$$

$$\text{s.t.} \quad \beta^\top \phi(s,a) \geqslant R(s,a) + \gamma \cdot \mathcal{P}^\pi \beta^\top \phi(s,a), \quad \forall (s,a) \in S \times A.$$

By Lemma 5, it is equivalent to solving

$$\min_{\beta \in \mathbb{R}^p} \quad \left\| Q^\pi - \beta^\top \phi \right\|_{1, \mu_0 \pi} \tag{19}$$

$$\text{s.t.} \quad \beta^\top \phi(s,a) \geqslant \mathcal{B}_\pi \left( \beta^\top \phi \right)(s,a), \quad \forall (s,a) \in S \times A.$$

We now define

$$(d^*, \beta^*) := \underset{d \geqslant 0}{\operatorname{argmax}} \ \underset{\beta}{\operatorname{argmin}} \ l(d, \beta),$$

$$\tilde{\beta} := \underset{\beta}{\operatorname{argmin}} \left\| Q^\pi - \beta^\top \phi \right\|_\infty,$$

$$\epsilon := \left\| Q^\pi - \tilde{\beta}^\top \phi \right\|_\infty,$$

and obtain from strong duality that

$$\mathbb{E}_{d^*}\left[r\left(s,a\right)\right] = (1-\gamma)\,\mathbb{E}_{\mu_0 \pi}\left[\left(\beta^*\right)^\top \phi\right].$$

Recall the fact $\mathcal{B}_\pi$ is a $\gamma$-contraction operator with the norm $\|\cdot\|_\infty$, and we have

$$\left\|\mathcal{B}_\pi\left(\tilde{\beta}^\top \phi\right) - Q^\pi\right\|_\infty \leqslant \gamma \left\|\tilde{\beta}^\top \phi - Q^\pi\right\|_\infty,$$

which implies

$$\mathcal{B}_\pi\left(\tilde{\beta}^\top \phi\right) \leqslant Q^\pi + \gamma\epsilon\mathbf{1}.$$

Now consider a new solution $\left(\tilde{\beta}^\top \phi - c\mathbf{1}\right)$, which must be in $\operatorname{span}\{\phi\}$ as $\mathbf{1} \in \operatorname{span}\{\phi\}$. Then,

$$\begin{aligned}
\mathcal{B}_\pi\left(\tilde{\beta}^\top \phi - c\mathbf{1}\right) &= \mathcal{B}_\pi\left(\tilde{\beta}^\top \phi\right) - \gamma c\mathbf{1} \\
&\leqslant Q^\pi + \gamma\epsilon\mathbf{1} - \gamma c\mathbf{1} \\
&\leqslant \tilde{\beta}^\top \phi + (1+\gamma)\,\epsilon\mathbf{1} - \gamma c\mathbf{1} \\
&= \tilde{\beta}^\top \phi - c\mathbf{1} + ((1-\gamma)\,c + (1+\gamma)\,\epsilon)\,\mathbf{1}.
\end{aligned}$$

Choose $c = -\left(1+\gamma\right)\epsilon/\left(1-\gamma\right)$, and the above implies $\mathcal{B}_\pi\left(\tilde{\beta}^\top \phi - c\mathbf{1}\right) \leqslant \tilde{\beta}^\top \phi - c\mathbf{1}$. Therefore, there exists some $\bar{\beta}$ such that

$$\bar{\beta}^\top \phi = \tilde{\beta}^\top \phi + \frac{1+\gamma}{1-\gamma}\epsilon\mathbf{1}.$$

Then, we can bound the approximation error

$$\begin{aligned}
\mathbb{E}_{d^*}\left[r\left(s,a\right)\right] - \rho_\pi &= \mathbb{E}_{d^*}\left[r\left(s,a\right)\right] - (1-\gamma)\,\mathbb{E}_{\mu_0 \pi}\left[Q^\pi\right] \\
&= (1-\gamma)\,\mathbb{E}_{\mu_0 \pi}\left[\left(\beta^*\right)^\top \phi\right] - (1-\gamma)\,\mathbb{E}_{\mu_0 \pi}\left[Q^\pi\right] \geqslant 0,
\end{aligned}$$

where the last inequality comes from the fact $(1-\gamma)\,\mathbb{E}_{\mu_0 \pi}\left[\left(\beta^*\right)^\top \phi\right]$ is the optimal value of a restricted feasible set within linearly representable $Q_\beta$.

On the other hand, we bound

$$\begin{aligned}
(1-\gamma)\,\mathbb{E}_{\mu_0 \pi}\left[\left(\beta^*\right)^\top \phi\right] - (1-\gamma)\,\mathbb{E}_{\mu_0 \pi}\left[Q^\pi\right] &= (1-\gamma)\left\|\left(\beta^*\right)^\top \phi - Q_\beta\right\|_{1,\mu_0 \pi} \\
&\leqslant (1-\gamma)\left\|\bar{\beta}^\top \phi - Q_\beta\right\|_{1,\mu_0 \pi} \\
&\leqslant (1-\gamma)\left\|\bar{\beta}^\top \phi - Q_\beta\right\|_\infty \\
&\leqslant (1-\gamma)\left(\left\|\bar{\beta}^\top \phi - \tilde{\beta}^\top \phi\right\|_\infty + \left\|Q^\pi - \tilde{\beta}^\top \phi\right\|_\infty\right) \\
&\leqslant (1-\gamma)\left(1 + \frac{1+\gamma}{1-\gamma}\right)\epsilon = 2\epsilon.
\end{aligned}$$

where the third inequality comes from the optimality of (19). ∎

**Justification of full-rank basis embedding.** The effect of full-rank basis embedding in the example in Section 3.1 can be justified straightforwardly. We consider the Lagrangian (17). If the $\phi \in \mathbb{R}^{|S||A| \times |S||A|}$ is full-rank, $\phi^{-1}$ exists. For arbitrary $Q \in \mathbb{R}^{|S||A| \times 1}$, there exists $\beta = \left(Q\phi^{-1}\right)^\top$, which means there is an one-to-one correspondence between $Q$ and $\beta$ in Lagrangian. Therefore, in finite state and action MDP, the Lagrangian is not affected by full-rank basis embedding, and therefore, the solution of full-rank basis embedding will be the same as the original LP.

# B  CoinDICE for Undiscounted and finite-horizon MDPs

In the main text, we consider the CoinDICE for infinite-horizon MDPs with discounted factor $\gamma < 1$. The algorithm can be generalized to undiscounted MDPs with $\gamma = 1$ and finite-horizon MDPs.

**Undiscounted MDP.** We have the dual form of the $Q$-LP as

$$\tilde{\rho}_\pi := \left\{ \max_{d:S\times A\to\mathbb{R}_+} \mathbb{E}_d[r\,(s,a)] \middle| \begin{array}{c} \int d\,(s,a)\,dsda = 1 \\ d\,(s,a) = \mathcal{P}_*^\pi d\,(s,a)\,,\forall\,(s,a)\in S\times A \end{array} \right\}. \tag{20}$$

Comparing with the (3), we have an extra normalization constraint. Specifically, if $d(s,a)$ is feasible, without the normalization constraint, $c\cdot d\,(s,a)$ will also be feasible for any $c > 0$. Therefore, the optimization could be unbounded.

By change-of-variable $\tau\,(s,a) = \frac{d^\pi(s,a)}{d^\mathcal{D}(s,a)}$ and feature embeddings of the stationary constraint in (20), we obtain

$$\tilde{\rho}_\pi := \left\{ \max_{\tau:S\times A\to\mathbb{R}_+} \mathbb{E}_{d^\mathcal{D}}[\tau\cdot r\,(s,a)] \middle| \begin{array}{c} \mathbb{E}_{d^\mathcal{D}}\,[\tau\,(s,a)] = 1 \\ \mathbb{E}_{d^\mathcal{D}}\,[\phi\,(s',a')\,(\tau\,(s',a') - \tau\,(s,a))] = 0 \end{array} \right\}. \tag{21}$$

Then, the CoinDICE confidence interval is achieved by applying the generalized empirical likelihood to (21), *i.e.*,

$$C_{n,\xi}^f := \left\{ \tilde{\rho}_\pi(w) = \max_{\tau\geqslant 0}\mathbb{E}_w\,[\tau\cdot r] \middle| \begin{array}{c} w\in\mathcal{K}_f \\ \mathbb{E}_w\,[\tau - 1] = 0 \\ \mathbb{E}_w\,[\bar{\Delta}\,(x;\tau,\phi)] = 0 \end{array} \right\}, \text{ with } \mathcal{K}_f := \left\{ \begin{array}{c} w\in\mathcal{P}^{n-1}\,(\hat{p}_n)\,, \\ D_f\,(w\|\hat{p}_n)\leqslant\frac{\xi}{n} \end{array} \right\}, \tag{22}$$

where $\bar{\Delta}\,(x;\tau,\phi) := \phi\,(s',a')\,(\tau\,(s',a') - \tau\,(s,a))$.

A similar argument of Section 3.3 for discounted MDPs can be applied to (22), resulting in the following confidence interval:

$$C_{n,\xi}^f = [l_n, u_n]$$

with

$$[l_n, u_n] = \left[ \min_{\beta\in\mathbb{R}^p,\nu}\max_{\tau\geqslant 0}\min_{w\in\mathcal{K}_f}\mathbb{E}_w\,[\ell\,(x;\tau,\beta,\nu)]\,, \quad \max_{\tau\geqslant 0}\min_{\beta\in\mathbb{R}^p,\nu}\max_{w\in\mathcal{K}_f}\mathbb{E}_w\,[\ell\,(x;\tau,\beta,\nu)] \right], \tag{23}$$

where $\ell\,(x;\tau,\beta,\nu) := \tau\cdot r + \beta^\top\Delta\,(x;\tau,\phi) + \nu - \nu\cdot\tau$.

**Remark (Normalization constraint):** Although in the discounted MDPs, there is no scaling issue, and thus the normalizaiton constraint is redudant, we still prefer to add the constraint in practice. It does not only bring the benefits in optimization, but also enforce the normalization explicitly and reduce the feasible set, leading to better statistical property.

**Finite-horizon MDP.** While we mainly focus on infinite-horizon MDPs with a discounted factor, the dual method can be adapted to finite-horizon settings straightforwardly. For example, we have the finite-horizon $d$-LP as

$$\max_{d_h(s,a):S\times A\to\mathbb{R}_+} \sum_{h=1}^H \mathbb{E}_{d_h}\,[r_h\,(s,a)] \tag{24}$$

$$\text{s.t.} \quad d_0\,(s,a) = \mu_0\,(s)\,\pi\,(a|s)\,, \tag{25}$$

$$d_{h+1}\,(s,a) = \mathcal{P}_*^\pi d_h\,(s,a)\,,\ \forall h\in\{1,\ldots,H\}\,. \tag{26}$$

Upon this finite-horizon formulation, we can derive the finite-horizon CoinDICE following the same technique, *i.e.*.

$$[l_n, u_n] =$$

$$\left[ \min_{w\in\mathcal{K}_f}\min_{\beta_{h=1}^H\in\mathbb{R}^p}\max_{\tau_{h=1}^H\geqslant 0}\mathbb{E}_w\left[\ell_H\left(x;\tau_{h=1}^H,\beta_{h=1}^H\right)\right]\,,\ \max_{w\in\mathcal{K}_f}\max_{\tau_{h=1}^H\geqslant 0}\min_{\beta_{h=1}^H\in\mathbb{R}^p}\mathbb{E}_w\left[\ell_H\left(x;\tau_{h=1}^H,\beta_{h=1}^H\right)\right] \right],$$

where $x := \left\{(s,a,r,s',a',h)_{h=1}^H\right\}$, $\ell_H\left(x;\tau_{h=1}^H,\beta_{h=1}^H\right) := \sum_{h=1}^H\tau_h r_h + \sum_{h=1}^H\beta_h^\top\Delta_h\,(x;\tau_h,\phi)$, and $\Delta_h\,(x;\tau_h,\phi) := \tau_h\,(s,a)\,\phi\,(s',a') - \tau_{h+1}\,(s',a')\,\phi\,(s',a')$.

# C CoinBandit

MDPs are strictly more general than multi-armed and contextual bandits. Therefore, our estimator can also be specialized accordingly for confidence interval estimation in bandit problems with slight modifications. Without loss of generality, we consider the contextual bandit setting, while the multi-armed bandits can be further reduced from contextual bandit.

Specifically, in the behavior-agnostic contextual bandit setting, the stationary distribution constraint in (5) is no long applicable in bandit setting. We rewrite the policy value as

$$
\begin{aligned}
\tilde{\rho}_\pi &:= \mathbb{E}_{s\sim\mu^{\mathcal{D}}, a\sim\pi(a|s)}\left[r\left(s,a\right)\right] \\
&= \left\{ \max_{\tau:S\times A\to\mathbb{R}_+} \mathbb{E}_{d^{\mathcal{D}}}\left[\tau\cdot r\left(s,a\right)\right] \,\middle|\, d^{\mathcal{D}}\cdot\tau=\mu^{\mathcal{D}}\pi, \mathbb{E}_{d^{\mathcal{D}}}\left[\tau\right]=1 \right\},
\end{aligned}
\tag{27}
$$

where we reload the $\mu^{\mathcal{D}}$ as the contextual distribution, which is unchanged for all policies, $d^{\mathcal{D}}\left(s,a\right) = \mu^{\mathcal{D}}\left(s\right)\pi_b\left(a|s\right)$, $\tau\left(s,a\right) := \frac{\mu^{\mathcal{D}}(s)\pi(a|s)}{\mu^{\mathcal{D}}(s)\pi_b(a|s)}$, and $\phi\left(s,a\right)$ denotes the feature mappings. We keep the normalization constraint to ensure the validation of density ratio empirically.

We apply the same technique to (27), leading to the *CoinBandit* confidence interval estimator

$$
C_{n,\xi}^f := \left\{ \tilde{\rho}_\pi(w) = \max_{\tau\geqslant 0}\mathbb{E}_w\left[\tau\cdot r\right] \,\middle|\, \begin{matrix} w\in\mathcal{K}_f, \mathbb{E}_w\left[\tau-1\right]=0 \\ \mathbb{E}_w\left[\blacktriangle\left(x;\tau,\phi\right)\right]=0 \end{matrix} \right\}, \quad \text{with } \mathcal{K}_f := \left\{ \begin{matrix} w\in\mathcal{P}^{n-1}\left(\widehat{p}_n\right), \\ D_f\left(w\|\widehat{p}_n\right)\leqslant\frac{\xi}{n} \end{matrix} \right\},
\tag{28}
$$

where the $x := \left(s,a,s',a'\right)$ is constructed by $s\sim\mu^{\mathcal{D}}\left(s\right), a\sim\pi\left(a|s\right)$ and $\left(s',a'\right)\sim d^{\mathcal{D}}$, and $\blacktriangle\left(x;\tau,\phi\right) := \phi\left(s,a\right)-\phi\left(s',a'\right)\cdot\tau\left(s',a'\right)$.

Similarly, the interval estimator in CoinBandit (28) can be calculated by solving a minimax optimization.

**Remark (Behavior-known contextual bandit):** When the behavior policy $\pi_b\left(a|s\right)$ is known, the solution to (27) can be computed in closed-form as $\tau\left(s,a\right)=\frac{\pi(a|s)}{\pi_b(a|s)}$. Then, the CoinBandit reduces to

$$
C_{n,\xi}^f := \left\{ \tilde{\rho}_\pi(w) = \mathbb{E}_w\left[\tau\cdot r\right] \,\middle|\, \begin{matrix} w\in\mathcal{K}_f, \\ \mathbb{E}_w\left[\tau-1\right]=0 \end{matrix} \right\}, \quad \text{with } \mathcal{K}_f := \left\{ \begin{matrix} w\in\mathcal{P}^{n-1}\left(\widehat{p}_n\right), \\ D_f\left(w\|\widehat{p}_n\right)\leqslant\frac{\xi}{n} \end{matrix} \right\}.
\tag{29}
$$

**Remark (Multi-armed bandit):** Furthermore, these estimators (28) and (29) can be further reduced for multi-armed bandit. Specifically, we set all $s$ equivalent, then, the $s$ becomes the dummy variable. The CoinBandit estimators (28) and (29) reduces for the off-policy evaluation in multi-armed bandit. If the action number is finite, we can use tabular representation for $\tau\left(a\right)$, eliminating the approximation error.

**Remark (Comparison to Karampatziakis et al. (2019)):** Karampatziakis et al. (2019) considers the off-policy contextual bandit confidence interval estimation. Although both CoinBandit and the estimator in Karampatziakis et al. (2019) share the same asymptotic coverage, there are significant differences:

- The estimator in Karampatziakis et al. (2019) is derived from empirical likelihood with reverse $KL$-divergence, while our CoinBandit is based on generalized empirical likelihood with arbitrary $f$-divergence.

- More importantly, compared to our CoinBandit, which is applicable for both *behavior-agnostic* and *behavior-known* off-policy setting, the estimator in Karampatziakis et al. (2019) is only valid for behavior-known setting.

- Computationally, the estimator in Karampatziakis et al. (2019) requires an extra statistics, *i.e.*,

$$
\left\{ \max_w \sum_{i=1}^n \log\left(nw_i\right) \,\middle|\, \mathbb{E}_w\left[\tau-1\right]=0, \ w\in\mathcal{K}_{-2\log(\cdot)} \right\},
$$

while such quantity is not required in CoinBandit, and thus saving the computational cost.

- Statistically, we provide finite sample complexity for CoinBandit in Theorem 4, while such sample complexity is not clear for Karampatziakis et al. (2019).

# D   Stochastic Confidence Interval Estimation

We analyze the properties of the optimization for the upper and lower bounds and derive the practical algorithm in this section.

### D.1 Upper and Lower Confidence Bounds

We first establish the distribution robust optimization representation of the confidence region:

**Lemma 6** *Let $\hat{\rho}_\pi(w) = \max_{\tau \geqslant 0} \min_{\beta \in \mathbb{R}^p} \mathbb{E}_w \left[ \tau \cdot r + \beta^\top \Delta(x; \tau, \phi) \right]$. The confidence region $C_{n,\xi}^f$ can be represented equivalently as*

$$C_{n,\xi}^f = \left\{ \hat{\rho}_\pi(w) \, \middle| \, w \in \mathcal{K}_f \right\}. \tag{30}$$

**Proof** For any $w \in \mathcal{K}_f$, we rewrite the optimization (8) by its Lagrangian, which will be an estimate of the policy value,

$$\hat{\rho}_\pi(w) = \max_{\tau \geqslant 0} \min_{\beta \in \mathbb{R}^p} \mathbb{E}_w \left[ \tau \cdot r + \beta^\top \Delta(x; \tau, \phi) \right]. \tag{31}$$

∎

Based on Lemma 6, we can formulate the upper and lower bounds:

**Theorem 3** *Denote the upper and lower confidence bounds of $C_{n,\xi}^f$ by $u_n$ and $l_n$, respectively:*

$$
\begin{aligned}
[l_n, u_n] &= \left[ \min_{w \in \mathcal{K}_f} \min_{\beta \in \mathbb{R}^p} \max_{\tau \geqslant 0} \mathbb{E}_w \left[ \ell(x; \tau, \beta) \right], \quad \max_{w \in \mathcal{K}_f} \max_{\tau \geqslant 0} \min_{\beta \in \mathbb{R}^p} \mathbb{E}_w \left[ \ell(x; \tau, \beta) \right] \right], \\
&= \left[ \min_{\beta \in \mathbb{R}^p} \max_{\tau \geqslant 0} \min_{w \in \mathcal{K}_f} \mathbb{E}_w \left[ \ell(x; \tau, \beta) \right], \quad \max_{\tau \geqslant 0} \min_{\beta \in \mathbb{R}^p} \max_{w \in \mathcal{K}_f} \mathbb{E}_w \left[ \ell(x; \tau, \beta) \right] \right],
\end{aligned}
$$

*where $\ell(x; \tau, \beta) := \tau \cdot r + \beta^\top \Delta(x; \tau, \phi)$. For any $(\tau, \beta, \lambda, \eta)$ that satisfies the constraints in (11), the optimal weights for upper and lower confidence bounds are*

$$w_l = f'_* \left( \frac{\eta - \ell(x; \tau, \beta)}{\lambda} \right) \quad \text{and} \quad w_u = f'_* \left( \frac{\ell(x; \tau, \beta) - \eta}{\lambda} \right),$$

*respectively. Therefore, the confidence bounds can be simplified as:*

$$
\begin{bmatrix} l_n \\ u_n \end{bmatrix} = \begin{bmatrix} \min_\beta \max_{\tau \geqslant 0, \lambda \geqslant 0, \eta} \mathbb{E}_{\mathcal{D}} \left[ -\lambda f_* \left( \frac{\eta - \ell(x; \tau, \beta)}{\lambda} \right) + \eta - \lambda \frac{\xi}{n} \right] \\ \max_{\tau \geqslant 0} \min_{\beta, \lambda \geqslant 0, \eta} \mathbb{E}_{\mathcal{D}} \left[ \lambda f_* \left( \frac{\ell(x; \tau, \beta) - \eta}{\lambda} \right) + \eta + \lambda \frac{\xi}{n} \right] \end{bmatrix}.
$$

**Proof** We first calculate the upper bound $u_n$ using Lemma 6:

$$
\begin{aligned}
u_n = \max_{w \in \mathcal{K}_f} \rho_\pi(w) &= \max_{w \in \mathcal{K}_f} \max_{\tau \geqslant 0} \min_{\beta \in \mathbb{R}^p} \mathbb{E}_w \left[ \tau \cdot r + \beta^\top \Delta(x; \tau, \phi) \right] \\
&= \max_{\tau \geqslant 0} \max_{w \in \mathcal{K}_f} \min_{\beta \in \mathbb{R}^p} \mathbb{E}_w \left[ \tau \cdot r + \beta^\top \Delta(x; \tau, \phi) \right] \tag{32} \\
&= \max_{\tau \geqslant 0} \min_{\beta \in \mathbb{R}^p} \max_{w \in \mathcal{K}_f} \mathbb{E}_w \left[ \tau \cdot r + \beta^\top \Delta(x; \tau, \phi) \right], \tag{33}
\end{aligned}
$$

where the switch between $\max_{w \in \mathcal{K}_f}$ and $\max_{\tau \geqslant 0}$ in (32) is immediate, (33) is due to the fact that the objective is concave w.r.t. $\beta$ and convex w.r.t. $w$ and $\tau$, separately.

We apply Lagrangian to the inner constrained optimization over $w$, leading to

$$
\begin{aligned}
u_n &= \max_{\tau} \min_{\beta, \lambda \geqslant 0, \eta} \max_{w \geqslant 0} \mathbb{E}_w \left[ \tau \cdot r + \beta^\top \Delta(x; \tau, \phi) \right] - \lambda \left( D_f(w \| \hat{p}_n) - \frac{\xi}{n} \right) + \eta \left( 1 - w^\top \mathbf{1} \right) \\
&= \max_{\tau \geqslant 0} \min_{\beta, \lambda \geqslant 0, \eta} \mathbb{E}_{\mathcal{D}} \left[ \lambda f_* \left( \frac{\tau \cdot r + \beta^\top \Delta(x; \tau, \phi) - \eta}{\lambda} \right) + \eta + \frac{\lambda \xi}{n} \right], \tag{34}
\end{aligned}
$$

where the last equation comes from the conjugate of $f$, and for any given $(\tau, \beta, \lambda, \eta)$, the optimal $w^*$ will be

$$w_u^* = f'_* \left( \frac{\tau \cdot r + \beta^\top \Delta(x; \tau, \phi) - \eta}{\lambda} \right).$$

The lower bound $l_n$ may be obtained in a similar fashion:

$$
\begin{aligned}
l_n = \min_{w \in \mathcal{K}_f} \rho(w; \pi) &= \min_{w \in \mathcal{K}_f} \max_{\tau \geqslant 0} \min_{\beta \in \mathbb{R}^p} \mathbb{E}_w \left[ \tau \cdot r + \beta^\top \Delta(x; \tau, \phi) \right] \\
&= \min_{w \in \mathcal{K}_f} \min_{\beta \in \mathbb{R}^p} \max_{\tau \geqslant 0} \mathbb{E}_w \left[ \tau \cdot r + \beta^\top \Delta(x; \tau, \phi) \right] \\
&= \min_{\beta \in \mathbb{R}^p} \min_{w \in \mathcal{K}_f} \max_{\tau \geqslant 0} \mathbb{E}_w \left[ \tau \cdot r + \beta^\top \Delta(x; \tau, \phi) \right] \\
&= \min_{\beta \in \mathbb{R}^p} \max_{\tau \geqslant 0} \min_{w \in \mathcal{K}_f} \mathbb{E}_w \left[ \tau \cdot r + \beta^\top \Delta(x; \tau, \phi) \right].
\end{aligned}
$$

Again, we consider the Lagrangian

$$
\begin{aligned}
l_n &= \min_{\beta \in \mathbb{R}^p} \max_{\tau \geqslant 0, \lambda \geqslant 0, \eta} \min_{w \geqslant 0} \mathbb{E}_w \left[ \tau \cdot r + \beta^\top \Delta(x; \tau, \phi) \right] + \lambda \left( D_f(w || \widehat{p}_n) - \frac{\xi}{n} \right) + \eta \left( 1 - w^\top \mathbf{1} \right) \\
&= \min_{\beta} \max_{\tau \geqslant 0, \lambda \geqslant 0, \eta} \mathbb{E}_\mathcal{D} \left[ -\lambda f_* \left( \frac{\eta - \left( \tau \cdot r + \beta^\top \Delta(x; \tau, \phi) \right)}{\lambda} \right) + \eta - \frac{\lambda \xi}{n} \right],
\end{aligned}
$$

and the optimal weight is

$$
w_l^* = f_*' \left( \frac{\eta - \left( \tau \cdot r + \beta^\top \Delta(x; \tau, \phi) \right)}{\lambda} \right).
$$

∎

## D.2 Closed-form Solution for Reweighting

We consider a few examples of $f$-divergences in Theorem 3, and show how the weights can be efficiently computed, for a given $\tau$ and $\beta$.

- $KL$-**divergence.** To satisfy the conditions in Assumption 1, we select $f(x) = 2x \log x$. Recall the property that for any convex function $f$ and any $\alpha > 0$, the conjugate function of $g(x) = \alpha f(x)$ is equal to $g_*(y) = \alpha f_*(y/\alpha)$. Let $f$ be the standard $f$-divergence function of $KL$-divergence $\mathrm{KL}(w || \widehat{p}_n)$, i.e., $f(x) = 2x \log x$. With $g_*'(y) = f_*'(y/\alpha)$, equation (12) implies that the following upper and lower bounds:

$$
w_l(x) = \exp \left( \frac{\eta_l - \ell(x; \tau, \beta)}{2\lambda} \right), \quad \eta_l = -\log \sum_{i=1}^n \exp \left( \frac{-\ell(x; \tau, \beta)}{2\lambda} \right)
$$

$$
w_u(x) = \exp \left( \frac{\ell(x; \tau, \beta) - \eta_u}{2\lambda} \right), \quad \eta_u = \log \sum_{i=1}^n \exp \left( \frac{\ell(x; \tau, \beta)}{2\lambda} \right).
$$

This can also be verified by plugging the $f(x) = 2x \log x$ into (12) and considering $w^\top \mathbf{1} = 1$.

- **Reverse KL-divergence.** With the f-divergence function $f(x) = -\log x$ for the reverse-KL divergence, one has the following upper and lower bounds:

$$
w_l(x) = \lambda \delta(\ell(x; \tau, \beta) > \eta_l) (\ell(x; \tau, \beta) - \eta_l)^{-1},
$$

$$
\sum_{i=1}^n \delta(\ell(x; \tau, \beta) > \eta_l) (\ell(x; \tau, \beta) - \eta_l)^{-1} = \frac{1}{\lambda},
$$

$$
w_u(x) = \lambda \delta(\eta_u > \ell(x; \tau, \beta)) (\eta_u - \ell(x; \tau, \beta))^{-1},
$$

$$
\sum_{i=1}^n \delta(\eta_u > \ell(x; \tau, \beta)) (\eta_u - \ell(x; \tau, \beta))^{-1} = \frac{1}{\lambda},
$$

where $\delta(a > b) = \begin{cases} 1 & \text{if } a > b \\ 0 & \text{otherwise} \end{cases}$. This is obtained by plugging the $f(x) = -\log x$ into (12) and considering $w^\top \mathbf{1} = 1$, $w \geqslant 0$ and KKT conditions on the dual variables for $w \geqslant 0$. Unfortunately the reverse KL-divergence does not satisfy the conditions in Assumption 1. Note that this is the standard f-divergence function for empirical likelihood maximization problem, we therefore also include it here for the sake of completeness.

- $\chi^2$**-divergence.** Notice that the standard f-divergence function, i.e., $f(x) = (x-1)^2$, of $\chi^2$-divergence $\chi^2(w\|\widehat{p}_n) := \mathbb{E}_{\widehat{p}_n}\left[\left(\frac{w}{\widehat{p}_n} - 1\right)^2\right]$ satisfies the conditions in Assumption 1. Consider the lower bound calculation. Leveraging the closed-form solution of the following $\ell_2$ projection problem onto the simplex space $w^\top \mathbf{1} = 1$ and $w \geqslant 0$ (Wang and Carreira-Perpiñán, 2013):

$$\arg\min_{w:w^\top \mathbf{1}=1,w\geqslant 0} \sum_{i=1}^n w_i \frac{\ell(x_i;\tau,\beta)}{\lambda} + \sum_{i=1}^n \frac{1}{\widehat{p}_{n,i}}(w_i - \widehat{p}_{n,i})^2$$

$$=\sqrt{\widehat{p}_{n,i}} \cdot \arg\min_{v:v^\top\sqrt{\widehat{p}_n}=1,v\geqslant 0} \sum_{i=1}^n \left(v_i - (1 - \frac{\ell(x_i;\tau,\beta)}{2\lambda})\cdot\sqrt{\widehat{p}_{n,i}}\right)^2, \text{ (here we let } v_i = \frac{w_i}{\sqrt{\widehat{p}_{n,i}}})$$

the lower bound $w_\ell(x)$ is given by (for any $i \in \{1,2,\ldots,n\}$)

$$w_\ell(x_i) = \sqrt{\widehat{p}_{n,i}} \cdot w^*(x_i)$$

$$= \sqrt{\widehat{p}_{n,i}} \cdot \left((1 - \frac{\ell(x_i;\tau,\beta)}{2\lambda})\cdot\sqrt{\widehat{p}_{n,i}} + \mathcal{G}_{\widehat{p}_n}\left((1 - \frac{\ell(x;\tau,\beta)}{2\lambda})\cdot\sqrt{\widehat{p}_{n,i}}\right)\right)_+,$$

where $\mathcal{G}_{\widehat{p}_n}(y) = \frac{1-\sum_{i=1}^{|\mathcal{S}_{\widehat{p}_n}|} y_i\cdot\sqrt{\widehat{p}_{n,i}}}{\sum_{i=1}^{|\mathcal{S}_{\widehat{p}_n}|}\widehat{p}_{n,i}}$, $\mathcal{S}_{\widehat{p}_n}$ is the set of indices in $\{1,\ldots,n\}$ in which any element $j$ satisfies $y_{(j)} + \frac{1}{\sum_{i=1}^j \widehat{p}_{n,i}}(1 - \sum_{i=1}^j y_{(i)}\cdot\sqrt{\widehat{p}_{n,i}}) > 0$. Here $y_{(i)}$ indicates the samples with the $i$-th largest element of $y$. Using analogous arguments, by replacing $\ell$ with $-\ell$ one can also define a similar solution for the upper bound $w_u(x)$. Now suppose $\widehat{p}_{n,i} = \frac{1}{n}, \forall i$. Then, we have

$$w_l(x_i) = \sqrt{\frac{1}{n}} \cdot \left((1 - \frac{\ell(x_i;\tau,\beta)}{2\lambda})\cdot\sqrt{\frac{1}{n}} + \mathcal{G}_{\frac{1}{n}}\left((1 - \frac{\ell(x;\tau,\beta)}{2\lambda})\cdot\sqrt{\frac{1}{n}}\right)\right)_+,$$

$$w_u(x_i) = \sqrt{\frac{1}{n}} \cdot \left((1 + \frac{\ell(x_i;\tau,\beta)}{2\lambda})\cdot\sqrt{\frac{1}{n}} + \mathcal{G}_{\frac{1}{n}}\left((1 + \frac{\ell(x;\tau,\beta)}{2\lambda})\cdot\sqrt{\frac{1}{n}}\right)\right)_+,$$

where $\mathcal{G}_{\frac{1}{n}}(y) = \frac{n-\sum_{i=1}^{|\mathcal{S}_{1/n}|} y_i\cdot\sqrt{n}}{|\mathcal{S}_{1/n}|}$, $\mathcal{S}_{\frac{1}{n}}$ is the set of indices in $\{1,\ldots,n\}$ in which any element $j$ satisfies $y_{(j)} + \frac{1}{j}(n - \sqrt{n}\sum_{i=1}^j y_{(i)}) > 0$. Here $y_{(i)}$ indicates the samples with the $i$-th largest element of $y$. This can also be verified by plugging the $f(x) = (x-1)^2$ into (12) and considering $w^\top \mathbf{1} = 1$ and $w \geqslant 0$. In fact, the above can be generalized to the Cressie-Read family with $f(x) = \frac{(x-1)^k - k(x-1) + k - 1}{k(k-1)}$.

- **Reverse $KL$-divergence.** With the $f$-divergence function $f(x) = -\log x$ for the reverse $KL$-divergence, one has the following upper and lower bounds:

$$w_l(x) = \lambda\delta(\ell(x;\tau,\beta) > \eta_l)(\ell(x;\tau,\beta) - \eta_l)^{-1},$$

$$\sum_{i=1}^n \delta(\ell(x;\tau,\beta) > \eta_l)(\ell(x;\tau,\beta) - \eta_l)^{-1} = \frac{1}{\lambda},$$

$$w_u(x) = \lambda\delta(\eta_u > \ell(x;\tau,\beta))(\eta_u - \ell(x;\tau,\beta))^{-1},$$

$$\sum_{i=1}^n \delta(\eta_u > \ell(x;\tau,\beta))(\eta_u - \ell(x;\tau,\beta))^{-1} = \frac{1}{\lambda},$$

where $\delta(a > b) = \begin{cases} 1 & \text{if } a > b \\ 0 & \text{otherwise} \end{cases}$. This is obtained by plugging the $f(x) = -\log x$ into (12) and considering $w^\top \mathbf{1} = 1$, $w \geqslant 0$ and KKT conditions on the dual variables for $w \geqslant 0$. Unfortunately the reverse $KL$-divergence does not satisfy the conditions in Assumption 1. Note that this is the standard $f$-divergence used in the vanilla empirical likelihood, we therefore also include it here for the sake of completeness.

### D.3 Practical Algorithm

In (13), we eliminate one level optimization, thus reduce the computational difficulty. Meanwhile, SGDA for (13) could benefit from the attractive finite-step convergence. However, as observed

**Algorithm 1** CoinDICE: estimating upper confidence bound using KL-divergence and function approximation.

---

**Inputs**: A target policy $\pi$, a desired confidence $1 - \alpha$, a finite sample dataset $\mathcal{D} := \{(s_0^{(j)}, a_0^{(j)}, s^{(j)}, a^{(j)}, r^{(j)}, s'^{(j)})\}_{j=1}^n$, optimizers $\mathcal{OPT}_\theta$, number of iterations $K, T$.

Set divergence limit $\xi := \frac{1}{2}\chi_1^{2,1-\alpha}$.
Initialize $\lambda \in \mathbb{R}$, $Q_{\theta_1} : S \times A \to \mathbb{R}$, $\zeta_{\theta_2} : S \times A \to \mathbb{R}$.
**for** $k = 1, \ldots, K$ **do**
  **for** $t = 1, \ldots, T$ **do**
    Sample from target policy $a_0^{(j)} \sim \pi(s_0^{(j)})$, $a^{(j)\prime} \sim \pi(s^{(j)\prime})$ for $j = 1, \ldots, n$.
    Compute loss terms:
$$\ell^{(j)} := (1-\gamma)Q_{\theta_1}(s_0^{(j)}, a_0^{(j)}) + \zeta_{\theta_2}(s^{(j)}, a^{(j)}) \cdot (-Q_{\theta_1}(s^{(j)}, a^{(j)}) + r^{(j)} + \gamma Q_{\theta_1}(s^{(j)\prime}, a^{(j)\prime}))$$

    Update $(\theta_1, \theta_2) \leftarrow \mathcal{OPT}_\theta(\mathcal{L}, \theta_1, \theta_2)$.
  **end for**
  Update $(w, \lambda)$ by (35) or (36)
  Compute loss $\mathcal{L} := \sum_{j=1}^n w^{(j)} \cdot \ell^{(j)}$.
**end for**

**Return** $\mathcal{L}$.

---

in Namkoong and Duchi (2016), when $\lambda$ approaches 0, SGDA for (13) may suffer from high variance. In this section, we consider two strategies to bypass such difficulty. We take the upper bound as an example, and the lower bound can be handled similarly:

- Instead of using the optimal weights (12), Namkoong and Duchi (2016) suggests to keep $(w, \lambda)$ in optimization to be updated simultaneously via gradients, *i.e.*, targeting on solving the Lagrangian (33) with SGDA directly. For example, with $KL$-divergence, this leads to the update of $w_u$ in the $t$-th iteration as

$$\tilde{w}^{(j)} = \exp\left(\eta_t \ell^{(j)}\right)\left(w^{(j)}\right)^{1-\eta_t \lambda}\left(\frac{1}{n}\right)^{\eta_t \lambda} \quad \text{and} \quad w_u = \frac{\tilde{w}^{(j)}}{\sum_j \tilde{w}^{(j)}}, \qquad (35)$$

  with stepsize $\eta_t$.

- The instability and high variance of solving (13) comes from unboundness of $w$ induced by arbitarry $\lambda$ during the optimization procedure. In other words, given a fixed $(\tau, \beta)$, if we can keep $w \in \mathcal{K}_f$ satisfied, *i.e.*,

$$w_u = \underset{KL(w||\hat{p}_n) \leqslant \frac{\xi}{n}}{\operatorname{argmax}} \langle w, \ell \rangle$$

$$\Rightarrow (w_u, \lambda^*) = \underset{w^\top \mathbf{1}=1, w \geqslant 0}{\operatorname{argmax}} \underset{\lambda \geqslant 0}{\operatorname{argmin}} \langle w, \ell \rangle - \lambda\left(KL\left(w||\hat{p}_n\right) - \frac{\xi}{n}\right)$$

$$\Rightarrow (w_u, \lambda^*) = \left\{\tilde{w}_{\lambda^*}^{(j)} := \exp\{\frac{\ell^{(j)}}{\lambda^*}\}; \quad w_{\lambda^*}^{(j)} := \frac{\tilde{w}_{\lambda^*}^{(j)}}{\sum \tilde{w}_{\lambda^*}^{(j)}} \text{ with } \sum_{j=1}^n w_{\lambda^*}^{(j)} \log w_{\lambda^*}^{(j)} = \frac{\xi}{n}\right\},$$
$$(36)$$

  the optimization will be stable.

Moreover, the major computation cost of optimization is updating the $w$, which is an $\mathcal{O}(n)$ operation. Therefore, we update $w$ less frequently, which corresponds to optimizing the equivalent form (10). Incorporating these techniques into SGDA, we obtain the algorithm in Algorithm 1.

**Remark (More regularization for stability):** Directly solving a Lagrangian for LP may induce instability, due to lack of curvature. To overcome such difficulty, the augmented Lagrangian method (ALM) (Rockafellar, 1974) is the natural choice. Directly applying the ALM will introduce the regularization $h\left(\mathbb{E}_{\hat{p}_n}\left[\Delta\left(x; \tau, \phi\right)\right]\right)$ where $h$ denotes some convex function with minimum at zero. Such regularization will not change the optimal solution $(\tau, \beta)$ in (11) and the value $[l_n, u_n]$.

The ALM introduces extra computational cost in optimization since the regularization involves empirical expectations inside a nonlinear function. We exploit alternative regularizations following the

spirit of ALM, while circumventing the computational difficulty. Recall the fact that the regularization on dual variable does not change the optimal solution (Nachum et al., 2019b, Theorem 4), *i.e.*

$$
\tau^*(s,a) \quad = \quad \left\{ \operatorname*{argmax}_{\tau \geqslant 0} \; \mathbb{E}_{d^{\mathcal{D}}} \left[ \tau \cdot r(s,a) \right] \big| \mathbb{E}_{d^{\mathcal{D}}} \left[ \Delta(x; \tau, \phi) \right] = 0 \right\} \tag{37}
$$

$$
= \quad \left\{ \operatorname*{argmax}_{\tau \geqslant 0} \; \mathbb{E}_{d^{\mathcal{D}}} \left[ \tau \cdot r(s,a) \right] - \alpha \mathbb{E}_p \left[ h(\tau) \right] \big| \mathbb{E}_{d^{\mathcal{D}}} \left[ \Delta(x; \tau, \phi) \right] = 0 \right\}, \tag{38}
$$

where $p$ is some distribution over $S \times A$.

We show the upper bound as an example, and the lower bound can be treated similarly. We have

$$
(w_u, \tau^*) \quad = \quad \operatorname*{argmax}_{w \in \mathcal{K}_f} \left\{ \operatorname*{argmax}_{\tau \geqslant 0} \; \mathbb{E}_w \left[ \tau \cdot r(s,a) \right] \big| \mathbb{E}_w \left[ \Delta(x; \tau, \phi) \right] = 0 \right\}
$$

$$
= \quad \operatorname*{argmax}_{w \in \mathcal{K}_f} \left\{ \operatorname*{argmax}_{\tau \geqslant 0} \; \mathbb{E}_w \left[ \tau \cdot r(s,a) \right] - \alpha \mathbb{E}_p \left[ h(\tau) \right] \big| \mathbb{E}_w \left[ \Delta(x; \tau, \phi) \right] = 0 \right\}, \tag{39}
$$

where the equality comes from Nachum et al. (2019b, Theorem 4) and the fact the regularization $\mathbb{E}_p \left[ h(\tau) \right]$ does not depend on $w$. Then, we can solve (39) alternatively for $(w_u, \tau^*)$ by Lagrangian,

$$
\max_{\tau \geqslant 0} \min_{\beta} \max_{w \in \mathcal{K}_f} \; \mathbb{E}_w \left[ \tau \cdot r(s,a) + \beta^\top \Delta(x; \tau, \phi) \right] - \alpha \mathbb{E}_p \left[ h(\tau) \right]. \tag{40}
$$

Although the optimal $\tilde{\beta}^*$ to (40) differs from $\beta^*$, $(w_u, \tau^*)$ are the same. Once we have the $(w_u, \tau^*)$, we can recover the original Lagrangian $\tilde{\rho}_\pi(w_u) = \mathbb{E}_{w_u} \left[ \tau \cdot r(s,a) \right]$, since $\mathbb{E}_{w_u} \left[ \beta^{*\top} \Delta(x; \tau^*, \phi) \right] = 0$ in the original Lagrangian $\mathbb{E}_w \left( \ell(x; \tau^*, \beta^*) \right)$ in (11) due to the KKT condition.

Comparing to the original ALM, the new regularization takes the advantage of ALM while keeps the original computational efficiency.

# E  Proofs for Statistical Properties

In this section, we provide the detailed proofs for the asymptotic coverage Theorem 2 and the finite-sample correction Theorem 4. For notation simplicity, we use sup, max and inf, min interchangeably. With a little abuse of notation, we use $\int$ as $\sum$ on discrete domain.

## E.1  Asymptotic Coverage

Theorem 2 follows from a result in Duchi et al. (2016). The following notation will be needed:

- $\ell(x; \tau, \beta) = (1 - \gamma) \beta^\top \phi(s_0, a_0) + \tau(s,a) \left( r(s,a) + \gamma \beta^\top \phi(s', a') - \beta^\top \phi(s,a) \right)$;

- $\|f\|_1 := \int |f(s,a)| \, d^{\mathcal{D}}(s,a) \, ds da$, and $\|\phi(s,a)\|_2 := \sqrt{\langle \phi, \phi \rangle}$;

- $\|f(s,a)\|_{L^2(d^{\mathcal{D}})} := \mathbb{E}_{d^{\mathcal{D}}} \left[ f^2(s,a) \right]^{\frac{1}{2}}$, $\mathcal{H} \subset L^2(d^{\mathcal{D}})$, we define $L^\infty(\mathcal{H})$ be the space of bounded linear functionals on $\mathcal{H}$ with $\|L_1 - L_2\|_{\mathcal{H}} := \sup_{h \in \mathcal{H}} |L_1 h - L_2 h|$ for $L_1, L_2 \in L^\infty(\mathcal{H})$;

- $p = \frac{dP}{d\mu}$, with a Lebesgue measure $\mu$, is the Radon-Nikodym derivative. Abusing notation a bit, we use $(D_f(P||Q), D(p||q))$, and $(\mathbb{E}_P[\cdot], \mathbb{E}_p(\cdot))$ interchangeably.

**Definition 7** *(Duchi et al., 2016,* **Hadamard directionally differentiability)** *Let $\mathcal{Q}$ be the space of signed measures bounded with norm $\|\cdot\|_{\mathcal{H}}$. The functional $T : \mathcal{P} \to \mathbb{R}$ is Hadamard directionally differentiable at $P \in \mathcal{P}$ tangentially to $B \subset \mathcal{Q}$ if for all $H \in B$, there exists $dT_p(H) \in \mathbb{R}$ such that for all convergent sequences $t_n \to 0$ and $\|H_n - H\|_{\mathcal{H}} \to 0$ that satisfies $P + t_n H_n \in \mathcal{P}$, the following holds*

$$
\frac{T(P + t_n H_n) - T(P)}{t_n} \to dT_P(H), \quad as \quad n \to \infty.
$$

*We say $T : \mathcal{P} \to \mathbb{R}$ has an* influence function *$T^1(x; P) \in \mathbb{R}$ if*

$$
dT_P(Q - P) := \int T^1(x; P) \, d(Q - P)(x),
$$

*and $\mathbb{E}_P \left[ T^1(x; P) \right] = 0$.*

We consider $f$ in $D_f$ satisfying the following assumption (Duchi et al., 2016),

**Assumption 1 (Smoothness of $f$-divergence)** *The function $f : \mathbb{R}_+ \to \mathbb{R}$ is convex, three times differentiable in a neighborhood of $1$, $f(1) = f'(1) = 0$ and $f''(1) = 2$.*[5]

Then, the following theorem, which slightly simplifies Duchi et al. (2016, Theorem 10), characterizes the asymptotic coverage of the general uncertainty estimation,

**Theorem 8 (General asymptotic coverage)** *Let Assumption 1 hold and $\mathcal{H} = \{h(x; \tau, \beta)\}$, where $h(x; \tau, \beta)$ is Lipschitz and the space of $(\tau, \beta)$ is compact. Denote $B \subset \mathcal{Q}$ be such that $\left\| \sqrt{n} \left( \widehat{P}_n - P_0 \right) - G \right\|_{\mathcal{H}} \to 0$ with $G \in B$. Assume $T : \mathcal{P} \to \mathbb{R}$ is Hadamard differentiable at $P_0$ tangentially to $B$ with influence function $T^1(\cdot; P_0)$ and $dT_P$ is defined and continuous on the whole $\mathcal{Q}$, then,*

$$\lim_{n \to \infty} \mathbb{P}\left( T(P_0) \in \left\{ T(P) : D_f(P \| P_n) \leqslant \frac{\xi}{n} \right\} \right) = \mathbb{P}\left( \chi_1^2 \leqslant \xi \right).$$

Denote the $T(P) = \max_{\tau \geqslant 0} \min_{\beta \in \mathbb{R}^p} \mathbb{E}_P [\ell(x; \tau, \beta)]$ by convexity-concavity, our proof for Theorem 2 will be mainly checking the conditions required by Theorem 8: **i)**, Lipschitz continuity of functions in $\mathcal{H}$, and **ii)** Hadamard differentiability of $T(P)$.

We first specify the regularity assumption for stationary distribution ratio:

**Assumption 2 (Stationary ratio regularity)** *The target stationary state-action correction rato is bounded: $\|\tau^*\|_\infty \leqslant C_\tau < \infty$, and $\tau^* \in \mathcal{F}_\tau$ where $\mathcal{F}_\tau$ is a convex, compact and bounded RKHS space with bounded kernel function $\|k((\cdot, \cdot), (s, a))\|_{\mathcal{F}_\tau} \leqslant K$.*

The bounded ratio component of Assumption 2 is a standard assumption used in Nachum et al. (2019a); Zhang et al. (2020a); Uehara et al. (2019). The latter part regarding $\mathcal{F}_\tau$ is required for the existence of solutions. In fact, the RKHS assumption $\mathcal{F}_\tau$ is already quite flexible, and it includes deep neural networks by adopting the neural tangent kernels (Arora et al., 2019).

With Assumption 2, we can immediately obtain

$$T(P) = \max_{\tau \in \mathcal{F}_\tau} \min_{\beta \in \mathbb{R}^p} \mathbb{E}_P [\ell(x; \tau, \beta)] = \min_{\beta \in \mathbb{R}^p} \max_{\tau \in \mathcal{F}_\tau} \mathbb{E}_P [\ell(x; \tau, \beta)]$$

by the minimax theorem (Ekeland and Temam, 1999, Proposition 2.1). By this equivalence, we will focus on the $\min$-$\max$ form.

Since $r \in [0, R_{\max}]$, one has for every $\pi$ that $Q^\pi \leqslant R_{\max}/(1 - \gamma)$. Therefore, it is reasonable to assume the following regularity conditions for $\phi$:

**Assumption 3 (Embedding feature regularity)** *There exist some finite constants $C_\beta$ and $C_\phi$, such that $\|\beta\|_2 \leqslant C_\beta$, $\|\phi\|_2 \leqslant C_\phi$. Moreover, $\phi(s, a)$ is $L_\phi$-Lipschitz continuous.*

This assumption implies $\|\beta^\top \phi\|_\infty \leqslant \|\beta\|_2 \|\phi\|_2 \leqslant C_\beta C_\phi$ and Lipschitz continuity of $\beta^\top \phi(s, a)$. We define $\mathcal{F}_\beta := \{\beta | \|\beta\|_2 \leqslant C_\beta\}$.

**Lemma 9 (Lipschitz continuity)** *Under Assumptions 2 and 3, function $\ell$ satisfies $\|\ell(x; \tau, \beta)\|_\infty \leqslant M$ and is $C_\ell$-Lipschitz in $(\tau, \beta)$, for some proper finite constants $M$ and $C_\ell$.*

**Proof** We first show the boundedness claim. By the definition of $\ell(x; \tau, \beta)$, one has

$$
\begin{aligned}
&\|\ell(x; \tau, \beta)\|_\infty \\
&\leqslant (1 - \gamma) \|\beta^\top \phi\|_\infty + \|\tau(s, a)(r(s, a) + \gamma \beta^\top \phi(s', a') - \beta^\top \phi(s, a))\|_\infty \\
&\leqslant (1 - \gamma) \|\beta^\top \phi\|_\infty + \|\tau(s, a)\|_\infty (r(s, a) + \gamma \beta^\top \phi(s', a') - \beta^\top \phi(s, a)) \\
&\leqslant (1 - \gamma) C_\beta C_\phi + C_\tau (R_{\max} + (1 + \gamma) C_\beta C_\phi) \\
&= (C_\tau + 1)(1 - \gamma) C_\beta C_\phi + C_\tau R_{\max} := M.
\end{aligned}
$$

We equip $\mathcal{F}_\tau \times \mathcal{F}_\beta$ with the norm

$$\|(\tau, \beta)\| := \|\tau\|_{\mathcal{F}_\tau} + \|\beta\|_2, \tag{41}$$

Then, we show the Lipschitz continuity of $\ell(x; \tau, \beta)$ in $(\tau, \beta)$,

$$
\begin{aligned}
&|\ell(x; \tau_1, \beta_1) - \ell(x; \tau_2, \beta_2)| \\
&\leqslant \ (1-\gamma) \left| \phi(s_0, a_0)^\top (\beta_1 - \beta_2) \right| + \left| \tau_2(s, a)(\beta_1 - \beta_2)^\top (\gamma \phi(s', a') + \phi(s, a)) \right| \\
&\quad + \left| (\tau_1(s, a) - \tau_2(s, a)) \left( r(s, a) + \gamma \beta_1^\top \phi(s', a') - \beta_1^\top \phi(s, a) \right) \right| \\
&\leqslant \ (1-\gamma)((2+\gamma)C_\phi + C_\tau) \|\beta_1 - \beta_2\|_2 + (R_{\max} + (1+\gamma)C_\phi C_\beta) |\tau_1(s, a) - \tau_2(s, a)|, \\
&\leqslant \ (1-\gamma)((2+\gamma)C_\phi + C_\tau) \|\beta_1 - \beta_2\|_2 + (R_{\max} + (1+\gamma)C_\phi C_\beta) K \|\tau_1 - \tau_2\|_{\mathcal{F}_\tau}, \\
&\leqslant \ C_\ell \left( \|\beta_1 - \beta_2\|_2 + \|\tau_1 - \tau_2\|_{\mathcal{F}_\tau} \right),
\end{aligned}
$$

which implies the $\ell(x; \tau, \beta)$ is $C_\ell$-Lipschitz continuous with

$$C_\ell := \max \left\{ (1-\gamma)((2+\gamma)C_\phi + C_\tau), (1+\gamma)C_\phi C_\beta) K \right\}. \qquad \blacksquare$$

We now check the Hadamard directional differentiability of $T(P)$. The following proof largely follows Duchi et al. (2016); Römisch (2014).

**Lemma 10 (Hadamard Differentiability)** *Under Assumptions 2 and 3, the functional $T(P) = \min_{\beta \in \mathcal{F}_\beta} \max_{\tau \in \mathcal{F}_\tau} \mathbb{E}_P[\ell(x; \tau, \beta)]$ is Hadamard directionally differentiable on $\mathcal{P}$ tangentially to $B(\mathcal{H}, P_0) \subset L^\infty(\mathcal{H})$ with derivative*

$$dT_P(H) := \int \ell(x; \tau^*, \beta^*) \, dH(x),$$

*where $(\beta^*, \tau^*) = \mathrm{argmin}_{\beta \in \mathcal{F}_\beta} \, \mathrm{argmax}_{\tau \in \mathcal{F}_\tau} \mathbb{E}_{P_0}[\ell(x; \tau, \beta)]$.*

**Proof** For convenience, we define

$$\tilde{H}(\tau, \beta) := \int \ell(x; \tau, \beta) \, dH(x),$$

where $H$ is associated with a measure in $\mathcal{Q}$.

We first show the upper bound convergence. For $H_n \in B(\mathcal{H}, P_0)$ with $\|H_n - H\|_{\mathcal{H}} \to 0$, for any sequence $t_n \to 0$, we have

$$
\begin{aligned}
&T(P_0 + t_n H_n) - T(P_0) \\
&= \ \min_{\beta \in \mathcal{F}_\beta} \max_{\tau \in \mathcal{F}_\tau} \left( \mathbb{E}_{P_0}[\ell(x; \tau, \beta)] + t_n \tilde{H}_n(\tau, \beta) \right) - \min_{\beta \in \mathcal{F}_\beta} \max_{\tau \in \mathcal{F}_\tau} \mathbb{E}_{P_0}[\ell(x; \tau, \beta)] \\
&\leqslant \ \max_{\tau \in \mathcal{F}_\tau} \left( \mathbb{E}_{P_0}[\ell(x; \tau, \beta^*)] + t_n \tilde{H}_n(\tau, \beta^*) \right) - \mathbb{E}_{P_0}[\ell(x; \tau, \beta^*)] \\
&\leqslant \ \max_{\tau \in \mathcal{F}_\tau} t_n \tilde{H}_n(\tau, \beta^*).
\end{aligned}
$$

Denote $\tau_n^* = \mathrm{argmax}_{\tau \in \mathcal{F}_\tau} \tilde{H}_n(\tau, \beta^*)$, by definition, we have

$$\max_{\tau \in \mathcal{F}_\tau} \tilde{H}_n(\tau, \beta^*) - \max_{\tau \in \mathcal{F}_\tau} \tilde{H}(\tau, \beta^*) \leqslant \tilde{H}_n(\tau_n^*, \beta^*) - \tilde{H}(\tau_n^*, \beta^*) \leqslant \left\| \tilde{H}_n - \tilde{H} \right\|_{\mathcal{H}} \to 0.$$

Therefore, we obtain

$$\limsup_n \frac{1}{t_n} \left( T(P_0 + t_n H_n) - T(P_0) \right) \leqslant \tilde{H}(\tau^*, \beta^*).$$

For the lower bound part, we have

$$
\begin{aligned}
&T(P_0 + t_n H_n) \\
&= \ \min_{\beta \in \mathcal{F}_\beta} \left\{ \max_{\tau \in \mathcal{F}_\tau} \left( \mathbb{E}_{P_0}[\ell(x; \tau, \beta)] + t_n \tilde{H}_n(\tau, \beta) \right) \right\} \\
&= \ \min_{\beta \in \mathcal{F}_\beta} \left\{ \mathbb{E}_{P_0}[\ell(x; \tau_n(\beta), \beta)] + t_n \left( \tilde{H}_n(\tau_n(\beta), \beta) - \tilde{H}(\tau_n(\beta), \beta) \right) + t_n \tilde{H}(\tau_n(\beta), \beta) \right\} \\
&\leqslant \ \min_{\beta \in \mathcal{F}_\beta} \left\{ \mathbb{E}_{P_0}[\ell(x; \tau_n(\beta), \beta)] + t_n \left\| \tilde{H}_n - \tilde{H} \right\|_{\mathcal{H}} + t_n \left\| \tilde{H} \right\|_{\mathcal{H}} \right\} \\
&\leqslant \ \min_{\beta \in \mathcal{F}_\beta} \mathbb{E}_{P_0}[\ell(x; \tau_n(\beta), \beta)] + \mathcal{O}(1) \cdot t_n,
\end{aligned}
$$

where $\tau_n(\beta) = \operatorname{argmax}_{\tau \in \mathcal{F}_\tau} \left( \mathbb{E}_{P_0}\left[\ell(x;\tau,\beta)\right] + t_n \tilde{H}_n(\tau,\beta) \right)$.

Denote the set of $\epsilon$-ball of solutions w.r.t. $P$ as

$$S_P(\epsilon) := \left\{ \beta \in \mathcal{F}_\beta : \max_{\tau \in \mathcal{F}_\tau} \mathbb{E}_P\left[\ell(x;\tau,\beta)\right] \leqslant \min_{\beta \in \mathcal{F}_\beta} \max_{\tau \in \mathcal{F}_\tau} \mathbb{E}_P\left[\ell(x;\tau,\beta)\right] + \epsilon \right\}.$$

Then, $\beta_n^* \in S_{P_0 + t_n H_n}(0)$ implies $\beta_n^* \in S_{P_0}(ct_n)$, which in turn implies the sequence of $\beta_n^*$ has a subsequence $\tilde{\beta}_m^*$ that converges to $\beta^* \in S_{P_0}(0)$.

It is straightforward to check the Lipschitz continuity of $\bar{\ell}(\beta) := \max_\tau \mathbb{E}\left[\ell(x;\tau,\beta)\right]$ as

$$\left| \bar{\ell}(\beta_1) - \bar{\ell}(\beta_2) \right|$$

$$\leqslant \quad (1-\gamma)\|\beta_1 - \beta_2\|_2 \, \mathbb{E}_{\mu_0\pi}\left[\|\phi_{s_0,a_0}\|\right]_2 + \left| \max_{\tau \in \mathcal{F}_\tau} \mathbb{E}\left[\tau \cdot r + \beta_1^\top \Delta\right] - \max_{\tau \in \mathcal{F}_\tau} \mathbb{E}\left[\tau \cdot r + \beta_2^\top \Delta\right] \right|$$

$$\leqslant \quad (1-\gamma)\|\beta_1 - \beta_2\|_2 \, \mathbb{E}_{\mu_0\pi}\left[\|\phi_{s_0,a_0}\|\right]_2 + \max_{\tau \in \mathcal{F}}\left| \mathbb{E}\left[\tau \cdot r + \beta_1^\top \Delta\right] - \mathbb{E}\left[\tau \cdot r + \beta_2^\top \Delta\right] \right|$$

$$\leqslant \quad (1-\gamma)\|\beta_1 - \beta_2\|_2 \, \mathbb{E}_{\mu_0\pi}\left[\|\phi_{s_0,a_0}\|\right]_2 + \max_{\tau \in \mathcal{F}}\left| \mathbb{E}\left[(\beta_1 - \beta_2)^\top \Delta\right] \right|$$

$$\leqslant \quad ((1-\gamma)C_\phi + C_\tau(1+\gamma)C_\phi)\|\beta_1 - \beta_2\|_2.$$

Therefore, with $\tilde{\beta}_n^* \to \beta^*$, we have

$$\lim_m \tilde{\ell}\left(\tilde{\beta}_m^*\right) = \min_\beta \tilde{\ell}(\beta) = T(P_0),$$

and due to the optimality, for any $m$,

$$\tilde{\ell}\left(\tilde{\beta}_m^*\right) \geqslant \min_\beta \tilde{\ell}(\beta).$$

$$T(P_0 + t_m H_m) - T(P_0)$$

$$\geqslant \quad \max_{\tau \in \mathcal{F}_\tau}\left\{ \mathbb{E}_{P_0}\left[\ell\left(x;\tau,\tilde{\beta}_m^*\right)\right] + t_n \tilde{H}_n\left(\tau,\tilde{\beta}_m^*\right) \right\} - \max_{\tau \in \mathcal{F}_\tau} \mathbb{E}_{P_0}\left[\ell\left(x;\tau,\tilde{\beta}_m^*\right)\right]$$

$$\geqslant \quad \mathbb{E}_{P_0}\left[\ell\left(x;\tau_m\left(\tilde{\beta}_m^*\right),\tilde{\beta}_m^*\right)\right] + t_n \tilde{H}_n\left(\tau_m\left(\tilde{\beta}_m^*\right),\tilde{\beta}_m^*\right) - \mathbb{E}_{P_0}\left[\ell\left(x;\tau_m\left(\tilde{\beta}_m^*\right),\tilde{\beta}_m^*\right)\right]$$

$$= \quad t_n \tilde{H}_n\left(\tau_m\left(\tilde{\beta}_m^*\right),\tilde{\beta}_m^*\right),$$

where $\tau_m\left(\tilde{\beta}_m^*\right) = \operatorname{argmax}_{\tau \in \mathcal{F}_\tau} \mathbb{E}_{P_0}\left[\ell\left(x;\tau,\tilde{\beta}_m^*\right)\right]$.

Since $\tilde{\beta}_m^* \to \beta^*$, we have $\tau_m\left(\tilde{\beta}_m^*\right) \to \tau^*$, and thus,

$$\left| \tilde{H}_n\left(\tau_m\left(\tilde{\beta}_m^*\right),\tilde{\beta}_m^*\right) - \tilde{H}(\tau^*,\beta^*) \right|$$

$$\leqslant \quad \left| \tilde{H}_n\left(\tau_m\left(\tilde{\beta}_m^*\right),\tilde{\beta}_m^*\right) - \tilde{H}\left(\tau_m\left(\tilde{\beta}_m^*\right),\tilde{\beta}_m^*\right) \right| + \left| \tilde{H}\left(\tau_m\left(\tilde{\beta}_m^*\right),\tilde{\beta}_m^*\right) - \tilde{H}(\tau^*,\beta^*) \right|$$

$$\leqslant \quad \left\| \tilde{H}_n - \tilde{H} \right\|_{\mathcal{H}} + \left| \tilde{H}\left(\tau_m\left(\tilde{\beta}_m^*\right),\tilde{\beta}_m^*\right) - \tilde{H}(\tau^*,\beta^*) \right| \to 0,$$

where we use $\ell(\tau,\beta;x)$ is Lipschitz continuous. Therefore, we obtain

$$\liminf_n \frac{1}{t_n}\left(T(P_0 + t_n H_n) - T(P_0)\right) \geqslant \tilde{H}(\tau^*,\beta^*).$$

∎

**Theorem 2 (Asymptotic coverage)** Under Assumptions 1, 2, and 3, if $\mathcal{D}$ contains *i.i.d.* samples and the optimal solution to the Lagrangian of (5) is unique, we have

$$\lim_{n\to\infty} \mathbb{P}\left(\rho_\pi \in C_{n,\xi}^f\right) = \mathbb{P}\left(\chi_{(1)}^2 \leqslant \xi\right). \tag{42}$$

Therefore, $C_{n,\chi_{(1)}^{2,1-\alpha}}^f$ is an asymptotic $(1-\alpha)$-confidence interval of the value of the policy $\pi$.

**Proof** The proof is to verify the conditions in Theorem 8 hold. By Lemma 6, we can rewrite

$$\mathbb{P}\left(\rho_\pi \in C_{n,\xi}^f\right) = \mathbb{P}\left(\rho_\pi \in \left\{\hat{\rho}_\pi(w) \mid w \in \mathcal{K}_f\right\}\right),$$

where, according to the boundedness assumption on $\beta$ in Assumption 3,

$$\hat{\rho}_\pi(w) = \max_{\tau \geqslant 0} \min_{\beta \in \mathcal{F}_\beta} \mathbb{E}_w \left[ \tau \cdot r + \beta^\top \Delta(x; \tau, \phi) \right] = \min_{\beta \in \mathcal{F}_\beta} \max_{\tau \geqslant 0} \mathbb{E}_w \left[ \tau \cdot r + \beta^\top \Delta(x; \tau, \phi) \right].$$

With Lemma 9 and Lemma 10, the conditions in Theorem 8 are satisfied. We apply Theorem 8 on the unique optimal solution $(\tau^*, \beta^*) = \operatorname{argmin}_{\beta \in \mathcal{F}_\beta} \operatorname{argmax}_{\tau \geqslant 0} \mathbb{E}_{P_0}[\ell(x; \tau, \beta)]$. We have $dT_P$ is a linear functional on the space of bounded measures and

$$dT_{P_0}(H) = \int \ell(x; \tau^*, \beta^*) \, dH(x),$$

with the canonical gradient given by $T^1(\cdot; P_0) = \ell(x; \tau^*, \beta^*) - \mathbb{E}_{P_0}[\ell(x; \tau^*, \beta^*)]$. ∎

## E.2  Finite-Sample Correction

The previous section considers the asymptotic coverage of CoinDICE. We now analyze the finite-sample effect for the estimator, for the special case $f(x) = (x-1)^2$. Thus, $D_f$ is the $\chi^2$-divergence.

Consider the optimization problem,

$$\max_{w \in \mathbb{R}^n} w^\top z, \quad \text{s.t.} \quad D_f(w \| \widehat{p}_n) \leqslant \frac{\xi}{n}, w \in \mathcal{P}^{n-1}(\widehat{p}_n). \tag{43}$$

The following result will be needed.

**Lemma 11** *(Namkoong and Duchi, 2017, **Theorem 1**) Let $Z \in [0, M]$ be a random variable, $\sigma^2 = Var(Z)$ and $s_n^2 = \mathbb{E}_{\widehat{p}_n}[Z^2] - \mathbb{E}_{\widehat{p}_n}[Z]^2$ as the population and sample variance of $Z$, respectively. For $\xi \geqslant 0$, we have*

$$\left[ \sqrt{\frac{\xi}{n} s_n^2} - \frac{M\xi}{n} \right]_+ \leqslant \max_w \left\{ \mathbb{E}_w[Z] \,|\, D_f(w \| \widehat{p}_n) \leqslant \frac{\xi}{n}, w \in \mathcal{P}^{n-1}(\widehat{p}_n) \right\} - \mathbb{E}_{\widehat{p}_n}[Z] \leqslant \sqrt{\frac{\xi}{n} s_n^2}.$$

*Moreover, for $n \geqslant \max \left\{ 2, \frac{M^2}{\sigma^2} \max\{4\sigma, 22\} \right\}$, with probability at least $1 - \exp\left(-\frac{3n\sigma^2}{5M^2}\right)$,*

$$\max_w \left\{ \mathbb{E}_w[Z] \,|\, D_f(w \| \widehat{p}_n) \leqslant \frac{\xi}{n}, w \in \mathcal{P}^{n-1}(\widehat{p}_n) \right\} = \mathbb{E}_{\widehat{p}_n}[Z] + \sqrt{\frac{\xi}{n} s_n^2}.$$

The follow is the symmetric version of Lemma 11, which can be obtained immediately by negating the random variable $Z$. For completeness, we give the proof below, which is adapted from Namkoong and Duchi (2017). Recall that the lower bound is obtained by solving the following:

$$\min_{w \in \mathbb{R}^n} w^\top z, \quad \text{s.t.} \quad D_f(w \| \widehat{p}_n) \leqslant \frac{\xi}{n}, \ w \in \mathcal{P}^{n-1}(\widehat{p}_n). \tag{44}$$

**Lemma 12 (Lower bound variance representation)** *Under the same conditions in Lemma 11, for $\xi \geqslant 0$, we have*

$$\left[ \sqrt{\frac{\xi}{n} s_n^2} - \frac{M\xi}{n} \right]_+ \leqslant \mathbb{E}_{\widehat{p}_n}[Z] - \min_w \left\{ \mathbb{E}_w[Z] \,|\, D_f(w \| \widehat{p}_n) \leqslant \frac{\xi}{n}, w \in \mathcal{P}^{n-1}(\widehat{p}_n) \right\} \leqslant \sqrt{\frac{\xi}{n} s_n^2}.$$

*Moreover, for $n \geqslant \max \left\{ 2, \frac{M^2}{\sigma^2} \max\{4\sigma, 22\} \right\}$, with probability at least $1 - \exp\left(-\frac{3n\sigma^2}{5M^2}\right)$,*

$$\min_w \left\{ \mathbb{E}_w[Z] \,|\, D_f(w \| \widehat{p}_n) \leqslant \frac{\xi}{n}, w \in \mathcal{P}^{n-1}(\widehat{p}_n) \right\} = \mathbb{E}_{\widehat{p}_n}[Z] - \sqrt{\frac{\xi}{n} s_n^2}.$$

**Proof** Denote $u = \frac{1}{n} - w$, we have $u^\top \mathbf{1} = 0$, and the optimization (44) can be written as

$$\bar{z} - \max_u u^\top (z - \bar{z}), \quad \text{s.t.} \quad \|u\|_2^2 \leqslant \frac{\xi}{n}, u^\top \mathbf{1} = 0, u \leqslant \frac{1}{n}, \tag{45}$$

with $\bar{z} = \frac{1}{n} \sum_{i=1}^n z_i$. Obviously, by the Cauchy-Schwartz inequality,

$$u^\top (z - \bar{z}) \leqslant \sqrt{\frac{\xi}{n}} \|z - \bar{z}\|_2,$$

and the equality holds if and only if

$$u_i = \frac{\sqrt{\xi}\,(z - \bar{z})}{n\,\|z - \bar{z}\|_2} = \frac{\sqrt{\xi}\,(z - \bar{z})}{n\sqrt{ns_n^2}}.$$

Given the constraint $u \leqslant \frac{1}{n}$, to achieve the maximum, one needs to ensure

$$\max_i \frac{\sqrt{\xi}\,(z - \bar{z})}{n\sqrt{ns_n^2}} \leqslant 1.$$

If this condition is satisfied, we have

$$\mathbb{E}_{\widehat{p}_n}[Z] - \min_w \left\{ \mathbb{E}_w\,(Z)\,|\,D_f\,(w\|\widehat{p}_n) \leqslant \frac{\xi}{n} \right\} \leqslant \sqrt{\frac{\xi}{n}\,s_n^2}.$$

Since $z \in [0, M]$, we have $|z_i - \bar{z}| \leqslant M$, to ensure the condition, we need $\frac{\xi M^2}{ns_n^2} \leqslant 1 \Leftrightarrow s_n^2 \geqslant \frac{\xi M^2}{n}$.
Otherwise, suppose $s_n^2 < \frac{\xi M^2}{n}$, or equivalently $\frac{\xi s_n^2}{n} < \frac{\xi^2 M^2}{n}$, then,

$$\min_w w^\top z \leqslant \mathbb{E}_{\widehat{p}_n}[z] - \left[ \sqrt{\frac{\xi}{n}\,s_n^2} - \frac{M\xi}{n} \right]_+.$$

For the high-probability statement, when $n \geqslant \max\left\{ 2, \frac{M^2}{\sigma^2}\max\{4\sigma, 22\} \right\}$, and the event $s_n^2 \geqslant \frac{3}{64}\sigma^2$
holds, $s_n^2 \geqslant \frac{\xi M^2}{n}$. Following Maurer and Pontil (2009, Theorem 10), one can bound that

$$\mathbb{P}\,(|s_n - \sigma| \leqslant a) \leqslant \exp\left( -\frac{na^2}{2M^2} \right).$$

Setting $a = \left( 1 - \frac{\sqrt{3}}{8} \right)\sigma$ completes the proof. ∎

With Lemma 11 and Lemma 12, we represent the confidence bounds with variance. We resort to an empirical Bernstein bound applied to the function space $\mathcal{F}$ with bounded function $h : \mathcal{X} \to [0, M]$, using empirical $\ell_\infty$-covering numbers, $\mathcal{N}_\infty\,(\mathcal{F}, \epsilon, n)$,

**Lemma 13** *(Maurer and Pontil, 2009,* **Theorem 6***) Let $n \geqslant \frac{8M^2}{t}$ and $t \geqslant \log 12$. Then, with probability at least $1 - 6\mathcal{N}_\infty\,(\mathcal{F}, \epsilon, 2n)\,e^{-t}$, for any $h \in \mathcal{F}$,*

$$\mathbb{E}\,[h] - \mathbb{E}_{\widehat{p}_n}[h] \leqslant \sqrt{\frac{18\,Var_{\widehat{p}_n}\,(h)\,t}{n}} + \frac{15Mt}{n} + 2\left( 1 + 2\sqrt{\frac{2t}{n}} \right)\epsilon.$$

**Theorem 4 (Finite-sample correction)** *Denote by $\mathcal{N}_\infty\,(\mathcal{F}_\tau, \epsilon, 2n)$ and $\mathcal{N}_\infty\,(\mathcal{F}_\beta, \epsilon, 2n)$ the $\ell_\infty$-covering numbers of $\mathcal{F}_\tau$ and $\mathcal{F}_\beta$ with $\epsilon$-ball on $2n$ empirical samples, respectively. Let $D_f$ be $\chi^2$-divergence. Under Assumptions 2 and 3, let $M := (C_\tau + 1)\,(1 - \gamma)\,C_\beta C_\phi + C_\tau R_{\max}$ and $C_\ell := \max\left\{ (1 - \gamma)\,((2 + \gamma)\,C_\phi + C_\tau, (1 + \gamma)\,C_\phi C_\beta)\,K \right\}$, then, we have*

$$\mathbb{P}\,(\rho_\pi \in [l_u - \zeta_n, u_n + \zeta_n]) \geqslant 1 - 12\mathcal{N}_\infty\,(\mathcal{F}_\tau, \epsilon, 2n)\,\mathcal{N}_\infty\,(\mathcal{F}_\beta, \epsilon, 2n)\,e^{-\frac{\xi}{18}},$$

*where $(l_n, u_n)$ are the solutions to (11), $\zeta_n = \frac{11M\xi}{6n} + 2\left( 1 + 2\sqrt{\frac{\xi}{9n}} \right)C_\ell \epsilon$ and $\xi = \chi_{(1)}^{2, 1-\alpha}$.*

*When the VC-dimensions of $\mathcal{F}_\tau$ and $\mathcal{F}_\beta$ (denoted by $d_{\mathcal{F}_\phi}$ and $d_{\mathcal{F}_\beta}$, respectively) are finite, we have*

$$\mathbb{P}\,(\rho_\pi \in [l_n - \kappa_n, u_n + \kappa_n]) \geqslant 1 - 12\exp\left( c_1 + 2\left( d_{\mathcal{F}_\tau} + d_{\mathcal{F}_\beta} - 1 \right)\log n - \frac{\xi}{18} \right),$$

*where $c_1 = 2c + \log d_{\mathcal{F}_\tau} + \log d_{\mathcal{F}_\beta} + \left( d_{\mathcal{F}_\tau} + d_{\mathcal{F}_\beta} - 1 \right)$, and $\kappa_n = \frac{11M\xi}{6n} + 2\frac{C_\ell M}{n}\left( 1 + 2\sqrt{\frac{\xi}{9n}} \right)$.*

**Proof** We focus on the upper bound, and the lower bound can be bounded in a similar way. Define

$$(\tau^*, \beta^*) := \operatorname*{argmax}_{\tau \in \mathcal{F}_\tau}\operatorname*{argmin}_\beta \mathbb{E}_{d^{\mathcal{D}}}\,[\ell\,(x; \tau, \beta)]$$

$$\left( \widehat{w}^*, \widehat{\tau}^*, \widehat{\beta}^* \right) := \operatorname*{argmax}_w\operatorname*{argmax}_{\tau \in \mathcal{F}_\tau}\operatorname*{argmin}_\beta \mathbb{E}_w\,[\ell\,(x; \tau, \beta)].$$

By definition and the optimality of $\beta^*$, we have

$$\rho_\pi = \mathbb{E}_{d^\mathcal{D}}\left[\ell\left(x;\tau^*,\beta^*\right)\right] \leqslant \mathbb{E}_{d^\mathcal{D}}\left[\ell\left(x;\tau^*,\hat{\beta}^*\right)\right]. \tag{46}$$

Applying Lemma 13 and the Lipschitz-continuity of $\ell\left(x;\tau,\beta\right)$ on $\mathcal{F}_\tau \times \mathcal{F}_\beta$, with probability at least $1 - 6\mathcal{N}_\infty\left(\mathcal{F}_\tau,\epsilon,2n\right)\mathcal{N}_\infty\left(\mathcal{F}_\beta,\epsilon,2n\right)e^{-t}$, we have

$$\mathbb{E}_{d^\mathcal{D}}\left[\ell\left(x;\tau^*,\hat{\beta}^*\right)\right]$$

$$\leqslant \mathbb{E}_{\widehat{p}_n}\left[\ell\left(x;\tau^*,\hat{\beta}^*\right)\right] + 3\sqrt{\frac{2Var_{\widehat{p}_n}\left(\ell\left(x;\tau^*,\hat{\beta}^*\right)\right)t}{n}} + \frac{15Mt}{n} + 2\left(1 + 2\sqrt{\frac{2t}{n}}\right)C_\ell\epsilon$$

$$\leqslant \max_{D_f(w\|\widehat{p}_n)\leqslant\frac{\xi}{n}}\mathbb{E}_w\left[\ell\left(x;\tau^*,\hat{\beta}^*\right)\right] - \left[\sqrt{\frac{\xi Var_{\widehat{p}_n}\left(\ell\left(x;\tau^*,\hat{\beta}^*\right)\right)}{n}} - \frac{M\xi}{n}\right]_+$$

$$+ 3\sqrt{\frac{2Var_{\widehat{p}_n}\left(\ell\left(x;\tau^*,\hat{\beta}^*\right)\right)t}{n}} + \frac{15Mt}{n} + 2\left(1 + 2\sqrt{\frac{2t}{n}}\right)C_\ell\epsilon$$

$$\leqslant \max_{D_f(w\|\widehat{p}_n)\leqslant\frac{\xi}{n}}\max_{\tau\in\mathcal{F}_\tau}\min_{\beta\in\mathcal{F}_\beta}\mathbb{E}_w\left[\ell\left(x;\tau,\beta\right)\right] + \frac{11}{6n}M\xi + 2\left(1 + 2\sqrt{\frac{2t}{n}}\right)C_\ell\epsilon$$

where the second equation comes from Lemma 11 and the third line comes from setting $t \leqslant \frac{\xi}{18}$ and the definition of $\hat{\beta}^*$. Combining this with (46), we may conclude that with probability at least $1 - 6\mathcal{N}_\infty\left(\mathcal{F}_\tau,\epsilon,2n\right)\mathcal{N}_\infty\left(\mathcal{F}_\beta,\epsilon,2n\right)e^{-\frac{\xi}{18}}$,

$$\rho_\pi \leqslant \max_{D_f(w\|\widehat{p}_n)\leqslant\frac{\xi}{n}}\max_{\tau\in\mathcal{F}_\tau}\min_{\beta\in\mathcal{F}_\beta}\mathbb{E}_w\left[\ell\left(x;\tau,\beta\right)\right] + \frac{11M\xi}{6n} + 2\left(1 + 2\sqrt{\frac{\xi}{9n}}\right)C_\ell\epsilon.$$

With the same strategy based on Lemma 12 and Lemma 13, we can also bound the finite-sample lower bound correction that with probability at least $1 - 6\mathcal{N}_\infty\left(\mathcal{F}_\tau,\epsilon,2n\right)\mathcal{N}_\infty\left(\mathcal{F}_\beta,\epsilon,2n\right)e^{-\frac{\xi}{18}}$,

$$\rho_\pi \geqslant \max_{D_f(w\|\widehat{p}_n)\leqslant\frac{\xi}{n}}\max_{\tau\in\mathcal{F}_\tau}\min_{\beta\in\mathcal{F}_\beta}\mathbb{E}_w\left[\ell\left(x;\tau,\beta\right)\right] - \frac{11M\xi}{6n} - 2\left(1 + 2\sqrt{\frac{\xi}{9n}}\right)C_\ell\epsilon.$$

The first part of the theorem is then proved.

For the second part, by van der Vaart and Wellner (1996, Theorem 2.6.7), one can bound $\mathcal{N}\left(\mathcal{F},\epsilon,2n\right) \leqslant c\text{VC}\left(\mathcal{F}\right)\left(\frac{16Mne}{\epsilon}\right)^{\text{VC}(\mathcal{F})-1}$ for some constant $c$. We set $\epsilon = \frac{M}{n}$ and denote $d_\mathcal{F} = \text{VC}\left(\mathcal{F}\right)$. Plugging this into the bound, we obtain

$$\mathbb{P}\left(\rho_\pi \in \left[l_n - \kappa, u_n + \kappa\right]\right) \geqslant 1 - 12\exp\left(c_1 + 2\left(d_{\mathcal{F}_\tau} + d_{\mathcal{F}_\beta} - 1\right)\log n - \frac{\xi}{18}\right),$$

where $c_1$ and $\kappa$ are as given in the theorem statement. ■

# F    Implementing Principles of Optimism and Pessimism

Based on the discussion in Section 5, the optimism and pessimism principles can be implemented by maximizing $u_\mathcal{D}\left(\pi\right)$ and $l_\mathcal{D}\left(\pi\right)$, respectively. In this section, we first calculate the gradient $\nabla_\pi u_\mathcal{D}\left(\pi\right)$ and $\nabla_\pi l_\mathcal{D}\left(\pi\right)$, and elaborate on the algorithm details.

Since we will optimize the policy $\pi$, we modify the confidence interval estimator in CoinDICE slightly, so that $\pi$ is an explicitly parameterized distribution. Concretely, we consider the samples $x := (s_0, s, a, r)$ with $s_0 \sim \mu_0(s)$, $(s, a, r, s') \sim d^\mathcal{D}$, which leads to the corresponding upper and lower bounds with

$$\tilde{\ell}\left(x;\tau,\beta,\pi\right) := \tau \cdot r + \beta^\top\tilde{\Delta}\left(x;\tau,\phi,\pi\right),$$

where $\tilde{\Delta}(x; \tau, \phi, \pi) = (1-\gamma)\mathbb{E}_{\pi(a_0|s_0)}[\phi(s_0, a_0)] + \gamma\mathbb{E}_{\pi(a'|s')}[\phi(s', a')\tau(s,a)] - \phi(s,a)\tau(s,a)$.

**Theorem 14** *Given optimal $(\beta_l^*, \tau_l^*, w_l^*)$ and $(\beta_u^*, \tau_u^*, w_u^*)$ for lower and upper bounds, respectively, the gradients of $l_{\mathcal{D}}(\pi)$ and $u_{\mathcal{D}}(\pi)$ can be computed as*

$$
\begin{bmatrix} \nabla_\pi l_{\mathcal{D}}(\pi) \\ \\ \nabla_\pi u_{\mathcal{D}}(\pi) \end{bmatrix} = \begin{bmatrix} \mathbb{E}_{w_l^*}\left[(1-\gamma)\mathbb{E}_{a_0\sim\pi}\left[\nabla_\pi\log\pi(a_0|s_0)\beta_l^{*\top}\phi(s_0,a_0)\right] + \\ \qquad \gamma\mathbb{E}_{a'\sim\pi(a'|s')}\left[\tau_l^*(s,a)\nabla_\pi\log\pi(a'|s')\beta_l^{*\top}\phi(s',a')\right]\right] \\ \mathbb{E}_{w_u^*}\left[(1-\gamma)\mathbb{E}_{a_0\sim\pi}\left[\nabla_\pi\log\pi(a_0|s_0)\beta_u^{*\top}\phi(s_0,a_0)\right] + \\ \qquad \gamma\mathbb{E}_{a'\sim\pi(a'|s')}\left[\tau_u^*(s,a)\nabla_\pi\log\pi(a'|s')\beta_u^{*\top}\phi(s',a')\right]\right] \end{bmatrix}
$$
(47)

**Proof** We focus on the computation of $\nabla_\pi u_{\mathcal{D}}(\pi)$ with the optimal $(\beta_u^*, \tau_u^*, w_u^*)$:

$$
\nabla_\pi u_{\mathcal{D}}(\pi) = \mathbb{E}_{w_u^*}\left[\nabla_\pi\tilde{\ell}(x; \tau, \beta)\right]
$$

$$
= (1-\gamma)\mathbb{E}_{w_u^*}\nabla_\pi\mathbb{E}_{a_0\sim\pi}\left[\beta_u^{*\top}\phi(s_0,a_0)\right] + \gamma\mathbb{E}_{w_u^*}\left[\tau_u^*(s,a)\nabla_\pi\mathbb{E}_{a'\sim\pi(a'|s')}\left[\beta_u^{*\top}\phi(s',a')\right]\right]
$$

$$
= (1-\gamma)\mathbb{E}_{w_u^*}\mathbb{E}_{a_0\sim\pi}\left[\nabla_\pi\log\pi(a_0|s_0)\beta_u^{*\top}\phi(s_0,a_0)\right]
$$
(48)

$$
+ \gamma\mathbb{E}_{w_u^*}\mathbb{E}_{a'\sim\pi(a'|s')}\left[\tau_u^*(s,a)\nabla_\pi\log\pi(a'|s')\beta_u^{*\top}\phi(s',a')\right].
$$
(49)

The case for the lower bound can be obtained similarly:

$$
\nabla_\pi l_{\mathcal{D}}(\pi) = (1-\gamma)\mathbb{E}_{w_l^*}\mathbb{E}_{a_0\sim\pi}\left[\nabla_\pi\log\pi(a_0|s_0)\beta_l^{*\top}\phi(s_0,a_0)\right]
$$
(50)

$$
+ \gamma\mathbb{E}_{w_l^*}\mathbb{E}_{a'\sim\pi(a'|s')}\left[\tau_l^*(s,a)\nabla_\pi\log\pi(a'|s')\beta_l^{*\top}\phi(s',a')\right].
$$

∎

Now, we are ready to apply the policy gradient upon $u_{\mathcal{D}}(\pi)$ or $l_{\mathcal{D}}(\pi)$ to implement the optimism for exploration or pessimism for safe policy improvement, respectively. We illustrate the details in Algorithm 2.

---

**Algorithm 2** CoinDICE-OPT: implementation of optimism/pessimism principle

---

**Inputs**: initial policy $\pi_0$, a desired confidence $1-\alpha$, a finite sample dataset $\mathcal{D} := \{x^{(j)} = (s_0^{(j)}, s^{(j)}, a^{(j)}, r^{(j)}, s'^{(j)})\}_{j=1}^n$, number of iterations $T$.

**for** $t = 1, \ldots, T$ **do**

    Estimate $(\beta_u^*, \tau_u^*, w_u^*)$ via Algorithm 1 for optimism. $\{(\beta_l^*, \tau_l^*, w_l^*)$ for pessimism.$\}$

    Sample $\{x^{(j)}\}_{j=1}^k \sim \mathcal{D}_t$, $a_0^{(j)} \sim \pi_t(s_0^{(j)})$, $a^{(j)\prime} \sim \pi_t(s^{(j)\prime})$ for $j = 1, \ldots, k$.

    Estimate the stochastic approximation to $\nabla_\pi u_{\mathcal{D}_t}(\pi_t)$ via (49). $\{\nabla_\pi l_{\mathcal{D}_t}(\pi_t)$ via (50) for pessimism.$\}$

    Natural policy gradient update: $\pi_{t+1} = \arg\min_\pi -\langle\pi, \nabla_\pi u_{\mathcal{D}_t}(\pi_t)\rangle + \frac{1}{\eta}KL(\pi||\pi_t)$.

    $\{\pi_{t+1} = \arg\min_\pi -\langle\pi, \nabla_\pi l_{\mathcal{D}_t}(\pi_t)\rangle + \frac{1}{\eta}KL(\pi||\pi_t)$ for pessimism.$\}$

    Collect samples $\mathcal{E} = \left\{x^{(j)} = (s_0, s, a, r, s')^{(j)}\right\}_{j=1}^m$ by executing $\pi_{t+1}$, $\mathcal{D}_{t+1} = \mathcal{D}_t \cup \mathcal{E}$.

    $\{$Skip the data collection step in offline setting.$\}$

**end for**

**Return** $\pi_T$.

---

## Footnotes

[5]That $f(1) = 0$ is required in the definition of f-divergence. If $f'(1) \neq 0$, one can "lift" it by $\bar{f}(t) = f(t) - f'(1)(t - 1)$ so that the new function satisfies $\bar{f}'(1) = 0$. $f''(1) = 2$ is assumed for easier calculation without loss of generality, as discussed in Duchi et al. (2016). For example, one can use $f(t) = 2x \log x - 2(x - 1)$ for modified KL-divergence, $f(t) = (x - 1)^2$ for $\chi^2$-divergence, and $f(t) = -\log x + (x - 1) - \frac{1}{2}(x - 1)^2$ for reverse KL-divergence.