[Reviews · NeurIPS 2020]

Review 1

Summary and Contributions: Update: Thanks for the detailed response! I felt the reviewers addressed my concerns adequately and as a result have not changed my score. ================== The paper aims to estimate confidence intervals for the off-policy estimate of a desired policy’s performance, even when the behavior policy is not available. To do so, the authors build upon recent works on off-policy estimation using duality. The broad gist of the proposed idea is similar to bootstrap--perturb input data and perform dual OPE procedure on it. However, unlike in the primal form, OPE in dual form has an inner optimization problem and so doing standard bootstrapping is too computationally expensive. To sideline this problem, the authors propose to (a) use function approximations to tackle the large set of SxA state-distribution constraints, and (b) choose a noise to control data perturbation such that minimizing/maximizing the dual objective with respect to the noise results in approximations to lower/upper bounds. Theoretically, under some assumptions on the function approximator, the proposed CIs are shown to be consistent and finite-sample coverage error is also established. Empirically, the proposed bound has more accurate coverage and also has tighter intervals than the existing methods. There are some aspects about the use of function approximators that could benefit from more analysis but overall the idea seems novel and has potential to be very useful.

Strengths: The main idea of the paper is really novel and clever. Confidence bounds are really important for certain RL applications, but the problems with importance sampling have limited their applicability to mostly short-horizon problems. The question of whether duality could be used to generate confidence bounds is a one that some researchers I know have wondered about, but they did not have any solid ideas of how to proceed. The approach proposed by the authors is by no means “obvious,” it is quite clever and likely required a great deal of thought. The proposed confidence bound has several very important features. One of the most important is that it is behavior-agnostic. This is extremely important in some applications because data otherwise comes either from humans or from deterministic scripts that implement some business logic, so there is no way to do importance sampling. Learning the behavior policy through imitation comes with many flaws. While the method’s coverage is imperfect when used with finite samples, the method comes pretty, and the empirical results are compelling. The consistency of the method with infinite samples is very interesting and surprising. The finite sample analysis is also a nice contribution. Lines 150-156 were really helpful in understanding the gist of the proposed method. It showed an interesting perspective from the point of view of bootstrapping.

Weaknesses: Confidence intervals depend on the choice of function approximator (to have a parameter configuration to satisfy the desired criteria) and also on the optimization procedure (to find that exact parameter configuration). Unlike prior bounds, which were non-parametric, the proposed bound is parametric and there is no definite way provided regarding how to select these parameters. Unfortunately, three such functions approximators are needed in practice, one for distribution ratio \tau, one for the Lagnrangian \beta, and other for the constraint embedding \phi. This makes the confidence intervals dependent on both the choice of neural-network architecture (#layers, #nodes/layer, activation function, etc) and the choice of optimization routine (step size, optimizer, initial distribution, etc.) used to find the saddle points. Further, the optimal design choices might vary from domain to domain, making it harder for the end-user to use this bound. However, like authors mention in Line 130-132, approximation errors are pretty much ignored in this paper, and the focus is on quantifying sampling error properly. Nonetheless, the idea presented is new, interesting, and principled; it can act as a foundation to stir future research along this direction to resolve above mentioned challenges. In Line 232, it is suggested how the noise term can be adjusted to obtain the CI with finite sample guarantee. Unfortunately, this depends on knowing the VC dimension of the function approximators, which is often itself is prohibitively hard. The method inherits one of the limitations of dual methods, that is, the proposed procedure assumes infinite horizon setting and it is not clear how to properly extend it to episodic settings. For the purpose of the experiments, termination with (1-\gamma) probability is used to get finite length trajectories.

Correctness: -------- Theory: I have not checked the proofs thoroughly. --------- Empirical: Line 231: Why isn’t the true value computed directly using a large number of monte-carlo samples from the simulator? How was the per-decision WIS computed? The one proposed by Precup et al. (the cited reference) is neither unbiased nor consistent (see section 3.9 for more discussion and section 3.10 for an improved estimator in the work by Thomas [1]). Further, the bounds used for baselines are principled only for unbiased estimators--WIS is biased. Although the work by Kuzborskij et al. [2] is recent, it might be better suited for getting CI with WIS estimators. Having these in future revisions might be more appropriate. It would also be interesting to see comparison to tighter methods such as the doubly robust estimator [3]. I suppose the bounds with unbiased estimators (e.g., per-decision importance sampling with Hoefdding’s) might have large CI widths, but having them in the appendix at least should be beneficial. While I think the content of the paper is more than adequate, and I suppose this would be more for future work, I would be very interested in seeing the estimator applied in other contexts, such as very long horizon problems or for off-policy learning from human experts. [1] Thomas, P. S. (2015). Safe reinforcement learning (Doctoral dissertation, University of Massachusetts Libraries). [2] Kuzborskij, I., Vernade, C., György, A., & Szepesvári, C. (2020). Confident Off-Policy Evaluation and Selection through Self-Normalized Importance Weighting. arXiv preprint arXiv:2006.10460. [3] Jiang, Nan, and Lihong Li. (2016.) "Doubly robust off-policy value evaluation for reinforcement learning." International Conference on Machine Learning.

Clarity: In general, I found the paper to be well written. One suggestion I have for the authors is to have a summary section in the start of the paper that provides a rough sketch of the entire pipeline of the proposed procedure with some additional hand-holding for the reader. Currently, there are many pieces to the entire idea and seeing the bigger picture is difficult at the first read. In Thm3, \lambda and \eta seem to be undefined in the main paper.

Relation to Prior Work: Yes.

Reproducibility: Yes

Additional Feedback: I was expecting to see empirical comparisons against CIs obtained using regression importance sampling [1] but It seems a little surprising that even without knowledge of the behavior policy the proposed method performs better than the baselines that assume “oracle” knowledge of behavior policies. Perhaps there is a way to make the proposed method even better by incorporating this knowledge, when available? [1] Hanna, J., Niekum, S., & Stone, P. (2019, May). Importance sampling policy evaluation with an estimated behavior policy. In International Conference on Machine Learning (pp. 2605-2613).


Review 2

Summary and Contributions: This paper studies the problem of behavior-agnostic off-policy evaluation in reinforcement learning, where the goal is to estimate a confidence interval on a target policy’s value, given only access to a static experience dataset collected by unknown behavior policies. The main contribution is a new approach – Confidence interval stationary distribution correction estimation (CoinDICE)--to obtaining confidence intervals for off-policy evaluation without relying on importance weighting. CoinDICE is derived by combining function space embedding and linear programming formation of off-policy evaluation problem. The validity of the method is proved theoretically in both asymptotic and finite-sample regimes. Moreover, empirically study is also performed to demonstrate that the proposed method can achieve tighter confidence interval estimates than existing methods. #-----------after rebuttal --------------- I have read the authors' rebuttal as well as other reviewer's comments. I think the author did a good job addressing most questions and comments. Hence I maintain my score.

Strengths: The method is well justified both theoretically and empirically.

Weaknesses: The experiment settings and testing domains are relatively simply.

Correctness: The problem formation and solution methods seem correct to me.

Clarity: The paper is well written

Relation to Prior Work: The relation to prior work is well discussed and this work is different from previous contributions.

Reproducibility: Yes

Additional Feedback: or the two-armed bandit problem, only one type of behavior policy is considered (choose the optimal arm with prob of 0.55), will the similar results be obtained with the behavior policy is changed? What are the behavior policies for FrozenLake, Taxi and Reacher?


Review 3

Summary and Contributions: The authors propose a new method CoinDICE to estimate confidence intervals for reinforcement learning.

Strengths: The authors developed a new method CoinDICE to estimate confidence interval for reinforcement learning. The algorithm builds on a few technical components, including a new feature embedded Q-LP, and a generalized empirical likelihood approach to confidence interval estimation. They analyzed the asymptotic coverage of CoinDICE’s estimate, and provided a finite-sample bound. They compared the new algorithm with several baselines and found it to be superior to them.

Weaknesses: The CoinDICE method seems to be complex to compute. The paper lacks discussion about how to solve optimization problems in CoinDICE and prove that the computation can be efficient.

Correctness: The claims seem to be correct

Clarity: The paper is well written. It lacks discussions about optimization problems in the proposed method.

Relation to Prior Work: The paper has a good introduction about previous works.

Reproducibility: No

Additional Feedback:

[Author Response · NeurIPS 2020]

We thank the reviewers for their close reading, detailed comments, and overall positive assessment. We address the
questions raised by each reviewer separately.

**Reviewer 1** Thanks for the appreciation and the concrete suggestions for paper refinement. We will include the
discussion and summary section for the final version.

• **VC dimension in finite-sample CI.** Since the estimator relies on some function approximator, the finite-sample
correction unavoidably relies on the complexity measures due to the union bound. We exploit the VC dimension due
to its generality. In fact, the bound can be improved by considering a data-dependent measure, *e.g.*, Rademacher
complexity, or by some function class dependent measure, *e.g.*, function norm in RKHS, for specific function
approximators. We will include these discussions in the final version.

• **Finite-horizon CoinDICE.** While we mainly focus on infinite-horizon MDPs with a discounted factor, the dual
method can be adapted to finite-horizon settings straightforwardly. For example, we have the finite-horizon $d$-LP as
$$\max_{d_h(s,a):S \times A \to \mathbb{R}_+} \sum_{h=1}^{H} \mathbb{E}_{d_h}[r_h(s,a)] \quad \text{s.t.} \quad d_{h+1}(s,a) = \mathcal{P}_*^\pi d_h(s,a), d_0(s,a) = \mu_0(s)\pi(a|s), \ \forall h \in \{1,\dots,H\}.$$
with the corresponding CoinDICE CI estimation as
$$[l_n, u_n] = \left[\min_{w \in \mathcal{K}_f} \min_{\beta_{h=1}^{H} \in \mathbb{R}^p} \max_{\tau_{h=1}^{H} \geqslant 0} \mathbb{E}_w\left[\ell_H\left(x; \tau_{h=1}^H, \beta_{h=1}^H\right)\right] \quad \max_{w \in \mathcal{K}_f} \max_{\tau_{h=1}^{H} \geqslant 0} \min_{\beta_{h=1}^{H} \in \mathbb{R}^p} \mathbb{E}_w\left[\ell_H\left(x; \tau_{h=1}^H, \beta_{h=1}^H\right)\right]\right],$$
where $x := \left\{(s,a,r,s',a',h)_{h=1}^H\right\}$, $\ell_H\left(x; \tau_{h=1}^H, \beta_{h=1}^H\right) := \sum_{h=1}^{H} \tau_h r_h + \sum_{h=1}^{H} \beta_h^\top \Delta_h(x; \tau_h, \phi)$, and
$\Delta_h(x; \tau_h, \phi) := \tau_h(s,a)\phi(s',a') - \tau_{h+1}(s',a')\phi(s',a')$.

• **Empirical comparison.** We compared with the WIS computed the same as in DualDICE/GenDICE papers and Liu
et al. (2018), which is found to work best out of all IS variants. In preliminary experiments we had assessed the
performance of other IS-based estimators as well as the clipped estimator from Thomas and Hoeffding's bound, but
found them all to yield worse empirical coverage. For simplicity we did not present it in the paper. We did not include
DR as a baseline, as the paper's focus is confidence interval estimation in infinite-horizon RL when marginalized
importance ratios are used. DR requires extra information about the model and/or value function, and will make the
comparison less clean. That said, an extension of CoinDICE to DR is indeed an interesting direction for future work.

• **True value** We did use the simulator to compute the groundtruth for the Bandit, FrozenLake, and Taxi environments.
However, for Reacher, we use a large ensemble of learned networks in order to account for issues in approximation
error.

• **Related work.** Regression IS: We focused comparisions with IS on scenarios where the behavior policy is known
for simplicity. Our comparisons show that CoinDICE's performance is better even when compared to these strong
baselines, suggesting that the IS ratio used in IS baselines likely leads to higher variance and thus looser bounds.
Self-normalized Importance Weighting (SN) (Kuzborskij et al. (2020)): Thanks for pointing out this related work,
which was released after the NeurIPS deadline. The major differences between CoinDICE and SN lies in three
aspects: **i)**, the CoinDICE is designed for RL setting while the SN is proposed for contextual bandit setting. It is
not clear how well a straightforward extension of SN to RL will work; **ii)**, CoinDICE is behavior-agnostic while
SN needs to know the behavior policy; **iii)**, CoinDICE is asymptotically pivotal, meaning that there are no hidden
quantities we need to estimate, while SN requires several unknown quantities, therefore, is not pivotal.
We will include these discussions and comparisons in our final version.

**Reviewer 2** We will provide further experimental details and empirical comparison in our final version, taking
advantage of the extra page.

• **More empirical performances with different behavior policies:**
With different behavior policies, the proposed CoinDICE achieves
similar performances. The comparison with 100 samples from
behavior policy choosing the optimal arm with probability 0.2 is
illustrated in Figure 1 due to space limitation. We will include the
complete comparison in our final version.

Figure 1: Additional comparison.

• **Behavior policies:** The behavior policy in FrozenLake is the optimal policy with 0.2 white noise, which reduces the
policy value dramatically, from 0.74 to 0.24. For the behavior policies in Taxi and Reacher, we follow the same
experiment setting for constructing the behavior policies to collect data as in DualDICE and Liu et al. (2018).

**Reviewer 4** **Computation of the CoinDICE:** Although the derivation of CoinDICE is nontrivial (which is part of our
major contributions), the optimal $w$ is given in Eq. 11, and instantiated for various $f$-divergence choices in Appendix
D.2. The final optimization problem is in a mini-max form (Eq. 14), which can be solved by stochastic gradient
methods. Full details are provided in Appendix D.3 with pseudocode (Algorithm 1). We will add more descriptions in
the final version using the extra page.

[Meta-Review · NeurIPS 2020]

With reviewer scores of (9, 7, 6) it seems extremely likely that this paper will be accepted. I generally agree with the reviewers that the approach is novel and clever, has a nice property of behavior agnostic (in terms of off-policy), and uses a duality approach to confidence bounds.